# Denoising Likelihood Score Matching for Conditional Score-based Data Generation

**Chen-Hao Chao**[*]
National Tsing Hua University

**Wei-Fang Sun**
National Tsing Hua University

**Bo-Wun Cheng**
National Tsing Hua University

**Yi-Chen Lo**
Mediatek Inc.

**Chia-Che Chang**
Mediatek Inc.

**Yu-Lun Liu**
Mediatek Inc.

**Yu-Lin Chang**
Mediatek Inc.

**Chia-Ping Chen**
Mediatek Inc.

**Chun-Yi Lee**[†]
National Tsing Hua University

## Abstract

Many existing conditional score-based data generation methods utilize Bayes' theorem to decompose the gradients of a log posterior density into a mixture of scores. These methods facilitate the training procedure of conditional score models, as a mixture of scores can be separately estimated using a score model and a classifier. However, our analysis indicates that the training objectives for the classifier in these methods may lead to a serious *score mismatch issue*, which corresponds to the situation that the estimated scores deviate from the true ones. Such an issue causes the samples to be misled by the deviated scores during the diffusion process, resulting in a degraded sampling quality. To resolve it, we formulate a novel training objective, called *Denoising Likelihood Score Matching* (DLSM) loss, for the classifier to match the gradients of the true log likelihood density. Our experimental evidences show that the proposed method outperforms the previous methods on both Cifar-10 and Cifar-100 benchmarks noticeably in terms of several key evaluation metrics. We thus conclude that, by adopting DLSM, the conditional scores can be accurately modeled, and the effect of the score mismatch issue is alleviated.

## 1 Introduction

Score-based generative models are probabilistic generative models that estimate score functions, i.e., the gradients of the log density for some given data distribution. As described in the pioneering work (Hyvärinen, 2005), the process of training score-based generative models is called *Score Matching* (SM), in which a score-based generative model is iteratively updated to approximate the true score function. Such a process often incurs heavy computational burdens, since the calculation of the score-matching objective involves the explicit computation of the partial derivatives of the score model during training. Therefore, a branch of study in this research domain (Vincent, 2011; Song et al., 2019) resorts to reformulating the score-matching objective to reduce the training cost. Among these works, the author in (Vincent, 2011) introduced the *Denoising Score-Matching* (DSM) method. This method facilitates the training process of score-based generative models, and thus lays the foundation for a number of subsequent researches. Recently, the authors in (Song & Ermon, 2019) proposed an unified framework based on DSM, and achieved remarkable performance on serval real-world datasets. Their success inspired several succeeding works (Song & Ermon, 2020; Ho et al., 2020; Song et al., 2021a;b; Dhariwal & Nichol, 2021), which together contribute to making score-based generative models an attractive choice for contemporary image generation tasks.

A favorable aspect of score-based generative models is their flexibility to be easily extended to conditional variants. This characteristic comes from a research direction that utilizes Bayes' theorem

---

[*]Work done during an internship at Mediatek Inc. E-mail: `lance_chao@gapp.nthu.edu.tw`
[†]Corresponding author. E-mail: `cylee@gapp.nthu.edu.tw`

to decompose a conditional score into a mixture of scores (Nguyen et al., 2017). Recent endeavors followed this approach and further extended the concept of conditional score-based models to a number of application domains, including colorization (Song et al., 2021b), inpainting (Song et al., 2021b), and source separation (Jayaram & Thickstun, 2020). In particular, some recent researchers (Song et al., 2021b; Dhariwal & Nichol, 2021) applied this method to the field of class-conditional image generation tasks, and proposed the *classifier-guidance* method. Different from the *classifier-guidance-free* method adopted by (Ho et al., 2021), they utilized a score model and a classifier to generate the posterior scores (i.e., the gradients of the log posterior density), with which the data samples of certain classes can be generated through the diffusion process. The authors in (Dhariwal & Nichol, 2021) showed that the classifier guidance method is able to achieve improved performance on large image generation benchmarks. In spite of their success, our analysis indicates that the conditional generation methods utilizing a score model and a classifier may suffer from a *score mismatch issue*, which is the situation that the estimated posterior scores deviate from the true ones. This issue causes the samples to be guided by inaccurate scores during the diffusion process, and may result in a degraded sampling quality consequently.

To resolve this problem, we first analyze the potential causes for the score mismatch issue through a motivational low-dimensional example. Then, we formulate a new loss function called *Denoising Likelihood Score-Matching* (DLSM) loss, and explain how it can be integrated into the current training method. Finally, we evaluate the proposed method under various configurations, and demonstrate its advantages in improving the sampling quality over the previous methods in terms of several evaluation metrics.

## 2 BACKGROUND

In this section, we introduce the essential background material for understanding the contents of this paper. We first introduce Langevin diffusion (Roberts & Tweedie, 1996; Roberts & Rosenthal, 1998) for generating data samples $\tilde{x} \in \mathbb{R}^d$ of a certain dimension $d$ from an unknown probability density function (pdf) $p(\tilde{x})$ using the score function $\nabla_{\tilde{x}} \log p(\tilde{x})$. Next, we describe Parzen density estimation (Parzen, 1962) and denoising score matching (DSM) (Vincent, 2011) for approximating $\nabla_{\tilde{x}} \log p(\tilde{x})$ with limited data samples. Finally, we elaborate on the conditional variant of the score function, i.e., $\nabla_{\tilde{x}} \log p(\tilde{x}|\tilde{y})$, and explains how it can be decomposed into $\nabla_{\tilde{x}} \log p(\tilde{x})$ and $\nabla_{\tilde{x}} \log p(\tilde{y}|\tilde{x})$ for some conditional variable $\tilde{y} \in \mathbb{R}^c$ of dimension $c$.

### 2.1 LANGEVIN DIFFUSION

Langevin diffusion (Roberts & Tweedie, 1996; Roberts & Rosenthal, 1998) can be used to generate data samples from an unknown data distribution $p(\tilde{x})$ using only the score function $\nabla_{\tilde{x}} \log p(\tilde{x})$, which is said to be well-defined if $p(\tilde{x})$ is everywhere non-zero and differentiable. Under the condition that $\nabla_{\tilde{x}} \log p(\tilde{x})$ is well-defined, Langevin diffusion enables $p(\tilde{x})$ to be approximated iteratively based on the following equation:

$$\tilde{x}_t = \tilde{x}_{t-1} + \frac{\epsilon^2}{2} \nabla_{\tilde{x}} \log p(\tilde{x}_{t-1}) + \epsilon z_t, \tag{1}$$

where $\tilde{x}_0$ is sampled from an arbitrary distribution, $\epsilon$ is a fixed positive step size, and $z_t$ is a noise vector sampled from a normal distribution $\mathcal{N}(\mathbf{0}, I_{d \times d})$ for simulating a $d$-dimensional standard Brownian motion. Under suitable regularity conditions, when $\epsilon \to 0$ and $T \to \infty$, $\tilde{x}_T$ is generated as if it is directly sampled from $p(\tilde{x})$ (Roberts & Tweedie, 1996; Welling & Teh, 2011). In practice, however, the data samples are generated with $\epsilon > 0$ and $T < \infty$, which violates the convergence guarantee. Although it is possible to use Metropolized algorithms (Roberts & Tweedie, 1996; Roberts & Rosenthal, 1998) to recover the convergence guarantee, we follow the assumption of the prior work (Song & Ermon, 2019) and presume that the errors are sufficiently small to be negligible when $\epsilon$ is small and $T$ is large.

The sampling process introduced in Eq. (1) can be extended to a time-inhomogeneous variant by making $p(\tilde{x}_t)$ and $\epsilon$ dependent on $t$ (i.e., $p_t(\tilde{x}_t)$ and $\epsilon_t$). Such a time-inhomogeneous variant is commonly adopted by recent works on score-based generative models (Song & Ermon, 2019; Song et al., 2021b), as it provides flexibility in controlling $p_t(\tilde{x}_t)$ and $\epsilon_t$. The experimental results in (Song & Ermon, 2019; Song et al., 2021b) demonstrated that such a time-inhomogeneous sampling process

can improve the sampling quality on real-world datasets. In Appendix A.6.1, we offer a discussion on the detailed implementation of such a time-inhomogeneous sampling process in this work.

## 2.2 Parzen Density Estimation

Given a true data distribution $p_{\text{data}}$, the empirical data distribution $p_0(\boldsymbol{x})$ is constructed by sampling $m$ independent and identically distributed data points $\{\boldsymbol{x}^{(i)}\}_{i=1}^m$, and can be represented as a sum of Dirac functions $\frac{1}{m}\sum_{i=1}^m \delta(\|\boldsymbol{x} - \boldsymbol{x}^{(i)}\|)$. Such a discrete data distribution $p_0(\boldsymbol{x})$ constructed from the dataset often violates the previous assumptions that everywhere is non-zero and is differentiable. Therefore, it is necessary to somehow adjust the empirical data distribution $p_0(\boldsymbol{x})$ before applying Langevin diffusion in such cases.

To deal with the above issue, a previous literature (Vincent, 2011) utilized Parzen density estimation to replace the Dirac functions with isotropic Gaussian smoothing kernels $p_\sigma(\tilde{\boldsymbol{x}}|\boldsymbol{x}) = \frac{1}{(2\pi)^{d/2}\sigma^d} e^{\frac{-1}{2\sigma^2}\|\tilde{\boldsymbol{x}} - \boldsymbol{x}\|^2}$ with variance $\sigma^2$. Specifically, Parzen density estimation enables the calculation of $p_\sigma(\tilde{\boldsymbol{x}}) = \frac{1}{m}\sum_{i=1}^m p_\sigma(\tilde{\boldsymbol{x}}|\boldsymbol{x}^{(i)})$. When $\sigma > 0$, the score function becomes well-defined and can thus be represented as the following:

$$\nabla_{\tilde{\boldsymbol{x}}} \log p_\sigma(\tilde{\boldsymbol{x}}) = \frac{\sum_{i=1}^m \frac{1}{\sigma^2}(\boldsymbol{x}^{(i)} - \tilde{\boldsymbol{x}})p_\sigma(\tilde{\boldsymbol{x}}|\boldsymbol{x}^{(i)})}{\sum_{i=1}^m p_\sigma(\tilde{\boldsymbol{x}}|\boldsymbol{x}^{(i)})}. \tag{2}$$

The proof for Eq. (2) is provided in Appendix A.2. This equation can be directly applied to Eq. (1) to generate samples with Langevin diffusion. Unfortunately, this requires summation over all $m$ data points during every iteration, preventing it from scaling to large datasets due to the rapid growth in computational complexity.

## 2.3 Denoising Score Matching

Score matching (SM) (Hyvärinen, 2005) was proposed to estimate the score function with a model $s(\tilde{\boldsymbol{x}}; \phi)$, parameterized by $\phi$. Given a trained score model $s(\tilde{\boldsymbol{x}}; \phi)$, the scores can be generated by a single forward pass, which reduces the computational complexity of Eq. (2) by a factor of $m$. To train such a score model, a straightforward approach is to use the Explicit Score-Matching (ESM) loss $L_{\text{ESM}}$, represented as:

$$L_{\text{ESM}}(\phi) = \mathbb{E}_{p_\sigma(\tilde{\boldsymbol{x}})}\left[\frac{1}{2}\|s(\tilde{\boldsymbol{x}}; \phi) - \nabla_{\tilde{\boldsymbol{x}}} \log p_\sigma(\tilde{\boldsymbol{x}})\|^2\right]. \tag{3}$$

This objective requires evaluating Eq. (2) for each training step, which also fails to scale well to large datasets. Based on Parzen density estimation, an efficient alternative, called Denoising Score-Matching (DSM) loss (Vincent, 2011), is proposed to efficiently calculate the equivalent loss $L_{\text{DSM}}$, expressed as:

$$L_{\text{DSM}}(\phi) = \mathbb{E}_{p_\sigma(\tilde{\boldsymbol{x}}, \boldsymbol{x})}\left[\frac{1}{2}\|s(\tilde{\boldsymbol{x}}; \phi) - \nabla_{\tilde{\boldsymbol{x}}} \log p_\sigma(\tilde{\boldsymbol{x}}|\boldsymbol{x})\|^2\right]. \tag{4}$$

where $\nabla_{\tilde{\boldsymbol{x}}} \log p_\sigma(\tilde{\boldsymbol{x}}|\boldsymbol{x})$ is simply $\frac{1}{\sigma^2}(\boldsymbol{x} - \tilde{\boldsymbol{x}})$. Since the computational cost of denoising score matching is relatively lower in comparison to other reformulation techniques (Hyvärinen, 2005; Song et al., 2019), it is extensively adopted in recent score-based generative models (Song & Ermon, 2019; 2020; Song et al., 2021b).

## 2.4 Conditional Score Decomposition via Bayes' Theorem

Score models can be extended to conditional models when conditioned on a certain label $\tilde{\boldsymbol{y}}$. Similar to $\tilde{\boldsymbol{x}}$, the smoothing kernels with variance $\tau^2$ can be applied on $\boldsymbol{y}$ to meet the requirement that the pdf is everywhere non-zero. Typically, $\tau$ is assumed to be sufficiently small so that $p_\tau(\tilde{\boldsymbol{y}}) \approx p(\boldsymbol{y})$. A popular approach adopted by researchers utilizes Bayes' theorem $p_{\sigma,\tau}(\tilde{\boldsymbol{x}}|\tilde{\boldsymbol{y}}) = p_{\sigma,\tau}(\tilde{\boldsymbol{y}}|\tilde{\boldsymbol{x}})p_\sigma(\tilde{\boldsymbol{x}})/p_\tau(\tilde{\boldsymbol{y}})$ to decompose the conditional score $\nabla_{\tilde{\boldsymbol{x}}} \log p_{\sigma,\tau}(\tilde{\boldsymbol{x}}|\tilde{\boldsymbol{y}})$ into a mixture of scores (Nguyen et al., 2017), which enables conditional data generation. Following the assumptions in the previous study (Song et al., 2021b), the decomposition can be achieved by taking the log-gradient on both sides of the equation, expressed as follows:

$$\nabla_{\tilde{\boldsymbol{x}}} \log p_{\sigma,\tau}(\tilde{\boldsymbol{x}}|\tilde{\boldsymbol{y}}) = \nabla_{\tilde{\boldsymbol{x}}} \log p_{\sigma,\tau}(\tilde{\boldsymbol{y}}|\tilde{\boldsymbol{x}}) + \nabla_{\tilde{\boldsymbol{x}}} \log p_\sigma(\tilde{\boldsymbol{x}}) - \underbrace{\nabla_{\tilde{\boldsymbol{x}}} \log p_\tau(\tilde{\boldsymbol{y}})}_{=0}, \tag{5}$$

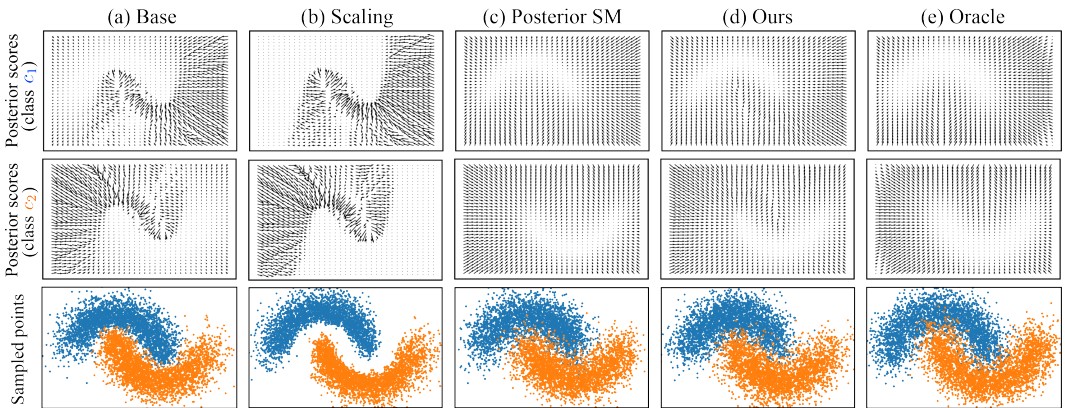

Figure 1: The visualized results on the *inter-twining moon* dataset. The plots presented in the first two rows correspond to the visualized vector fields for the posterior scores of the class $c_1$ (upper crescent) and $c_2$ (lower crescent), respectively. The plots in the third row are the sampled points. Different columns correspond to different experimental settings in Section 3. The detailed configurations are presented in Appendix A.6.2.

where $\nabla_{\tilde{\boldsymbol{x}}} \log p_{\sigma,\tau}(\tilde{\boldsymbol{x}}|\tilde{\boldsymbol{y}})$ is the posterior score, $\nabla_{\tilde{\boldsymbol{x}}} \log p_{\sigma,\tau}(\tilde{\boldsymbol{y}}|\tilde{\boldsymbol{x}})$ is the likelihood score, and $\nabla_{\tilde{\boldsymbol{x}}} \log p_\sigma(\tilde{\boldsymbol{x}})$ is the prior score. Base on Eq. (5), a conditional score model $\nabla_{\tilde{\boldsymbol{x}}} \log p(\tilde{\boldsymbol{x}}|\tilde{\boldsymbol{y}}; \phi, \theta)$ can be represented as the combination of the log-gradient of a differentiable classifier $p(\tilde{\boldsymbol{y}}|\tilde{\boldsymbol{x}}; \theta)$ and a prior score model $s(\tilde{\boldsymbol{x}}; \phi)$:

$$\nabla_{\tilde{\boldsymbol{x}}} \log p(\tilde{\boldsymbol{x}}|\tilde{\boldsymbol{y}}; \theta, \phi) = \nabla_{\tilde{\boldsymbol{x}}} \log p(\tilde{\boldsymbol{y}}|\tilde{\boldsymbol{x}}; \theta) + s(\tilde{\boldsymbol{x}}; \phi). \tag{6}$$

This formulation enables conditional data generation using a classifier $p(\tilde{\boldsymbol{y}}|\tilde{\boldsymbol{x}}; \theta)$ trained with cross-entropy loss $L_{\text{CE}}(\theta) \triangleq \mathbb{E}_{p_{\sigma,\tau}(\tilde{\boldsymbol{x}},\tilde{\boldsymbol{y}})}[-\log p(\tilde{\boldsymbol{y}}|\tilde{\boldsymbol{x}}; \theta)]$, and a score model $s(\tilde{\boldsymbol{x}}; \phi)$ trained with denoising score matching loss $L_{\text{DSM}}(\phi)$. Unfortunately, a few previous studies (Nguyen et al., 2017; Dhariwal & Nichol, 2021) have noticed that the approximation of $\nabla_{\tilde{\boldsymbol{x}}} \log p(\tilde{\boldsymbol{x}}|\tilde{\boldsymbol{y}}; \theta, \phi)$ is empirically inaccurate (referred to as the score mismatch issue in this work), and leveraged a scaling factor $\alpha > 0$ to adjust the likelihood score $\nabla_{\tilde{\boldsymbol{x}}} \log p(\tilde{\boldsymbol{y}}|\tilde{\boldsymbol{x}}; \theta)$. Such a scaling factor is a hyperparameter that empirically controls the amount of conditional information incorporated during the sampling process (see Appendix A.3). However, this usually causes the diversity of generated data samples to degrade noticeably (Dhariwal & Nichol, 2021), which is later discussed in Section 5.2.

In this work, we examine and investigate the score mismatch issue from a different perspective. We first offer a motivational example to show that the issue remains even with the scaling technique. Then, we theoretically formulate a new loss term that enables a classifier $p(\tilde{\boldsymbol{y}}|\tilde{\boldsymbol{x}}; \theta)$ to produce better likelihood score estimation.

## 3   ANALYSIS ON THE SCORE MISMATCH ISSUE

To further investigate the score mismatch issue, we first leverage a motivational experiment on the *inter-twining moon* dataset to examine the extent of the discrepancy between the estimated and true posterior scores. In this experiment, we consider five different methods to calculate the posterior scores denoted as (a)~(e):

- **(a) Base method.** The posterior scores are estimated using a score model $s(\tilde{\boldsymbol{x}}; \phi)$ trained with $L_{\text{DSM}}$ and a classifier $p(\tilde{\boldsymbol{y}}|\tilde{\boldsymbol{x}}; \theta)$ trained with $L_{\text{CE}}$, as described in Eq. (6).

- **(b) Scaling method.** The posterior scores are estimated in a similar fashion as method (a) except that the scaling technique (Dhariwal & Nichol, 2021) is adopted to scale the likelihood scores, i.e., $\nabla_{\tilde{\boldsymbol{x}}} \log p(\tilde{\boldsymbol{x}}|\tilde{\boldsymbol{y}}; \theta, \phi) = \alpha \nabla_{\tilde{\boldsymbol{x}}} \log p(\tilde{\boldsymbol{y}}|\tilde{\boldsymbol{x}}; \theta) + s(\tilde{\boldsymbol{x}}; \phi)$, where $\alpha > 0$.

- **(c) Posterior SM method.** The posterior scores for different class conditions, in this case, $c_1$ and $c_2$, are separately estimated using different score models $s(\tilde{\boldsymbol{x}}; \phi_1)$ and $s(\tilde{\boldsymbol{x}}; \phi_2)$ trained with $L_{\text{DSM}}$. Note that, since this method requires network architectures different from

Table 1: The experimental results on the inter-twining moon dataset. The quality of the sampled data for different methods are measured in terms of the precision and recall metrics. The errors of the score functions for different methods are measured using $\mathbb{E}_{U(\tilde{\boldsymbol{x}})}[D_P(\tilde{\boldsymbol{x}}, \tilde{\boldsymbol{y}})]$ and $\mathbb{E}_{U(\tilde{\boldsymbol{x}})}[D_L(\tilde{\boldsymbol{x}}, \tilde{\boldsymbol{y}})]$, where $U(\tilde{\boldsymbol{x}})$ is a uniform distribution over a two-dimensional subspace (i.e., the space of each subplot in Fig. 1), $\tilde{\boldsymbol{y}}$ represents the classes specified in the second row of the table, and $D_P$ and $D_L$ are defined in Eq. (7). The configurations and the hyperparameters for different experimental settings of this experiment are presented in Appendix A.6.2.

| | (a) Base | | (b) Scaling | | (c) Posterior SM | | (d) Ours | | (e) Oracle | |
|---|---|---|---|---|---|---|---|---|---|---|
| $\tilde{\boldsymbol{y}}$ | $c_1$ | $c_2$ | $c_1$ | $c_2$ | $c_1$ | $c_2$ | $c_1$ | $c_2$ | $c_1$ | $c_2$ |
| $S_P(\tilde{\boldsymbol{x}}, \tilde{\boldsymbol{y}})$ | $\nabla_{\tilde{\boldsymbol{x}}} \log p(\tilde{\boldsymbol{y}}\|\tilde{\boldsymbol{x}}; \theta) + s(\tilde{\boldsymbol{x}}; \phi)$ | | $\alpha\nabla_{\tilde{\boldsymbol{x}}} \log p(\tilde{\boldsymbol{y}}\|\tilde{\boldsymbol{x}}; \theta) + s(\tilde{\boldsymbol{x}}; \phi)$ | | $s(\tilde{\boldsymbol{x}}; \phi_1)$ | $s(\tilde{\boldsymbol{x}}; \phi_2)$ | $\nabla_{\tilde{\boldsymbol{x}}} \log p(\tilde{\boldsymbol{y}}\|\tilde{\boldsymbol{x}}; \theta) + s(\tilde{\boldsymbol{x}}; \phi)$ | | $\nabla_{\tilde{\boldsymbol{x}}} \log p_{\sigma,\tau}(\tilde{\boldsymbol{x}}\|\tilde{\boldsymbol{y}})$ | |
| $S_L(\tilde{\boldsymbol{x}}, \tilde{\boldsymbol{y}})$ | $\nabla_{\tilde{\boldsymbol{x}}} \log p(\tilde{\boldsymbol{y}}\|\tilde{\boldsymbol{x}}; \theta)$ | | $\alpha\nabla_{\tilde{\boldsymbol{x}}} \log p(\tilde{\boldsymbol{y}}\|\tilde{\boldsymbol{x}}; \theta)$ | | - | | $\nabla_{\tilde{\boldsymbol{x}}} \log p(\tilde{\boldsymbol{y}}\|\tilde{\boldsymbol{x}}; \theta)$ | | $\nabla_{\tilde{\boldsymbol{x}}} \log p_{\sigma,\tau}(\tilde{\boldsymbol{y}}\|\tilde{\boldsymbol{x}})$ | |
| $\mathbb{E}_{U(\tilde{\boldsymbol{x}})}[D_P(\tilde{\boldsymbol{x}}, \tilde{\boldsymbol{y}})]$ | 0.198 | 0.197 | 2.734 | 2.627 | 0.063 | 0.060 | 0.066 | 0.064 | 0.000 | 0.000 |
| $\mathbb{E}_{U(\tilde{\boldsymbol{x}})}[D_L(\tilde{\boldsymbol{x}}, \tilde{\boldsymbol{y}})]$ | 0.190 | 0.181 | 2.730 | 2.618 | - | - | 0.052 | 0.052 | 0.000 | 0.000 |
| Precision | 0.94 | 0.93 | 0.95 | 0.94 | 0.97 | 0.97 | 0.96 | 0.95 | 1.00 | 1.00 |
| Recall | 1.00 | 1.00 | 0.97 | 0.97 | 1.00 | 1.00 | 1.00 | 1.00 | 1.00 | 1.00 |

those used in methods (a), (b), and (d), it is only adopted for analytical purposes. For more detailed discussions about this method, please refer to Appendix A.4.

- **(d) Ours.** The posterior scores are estimated in a similar fashion as method (a) except that the classifier is trained with the proposed loss function $L_{\text{Total}}$, which is described in detail in Section 4.

- **(e) Oracle.** The oracle posterior scores are directly computed using the conditional variant of Eq. (2), which is detailed in Appendix A.6.2.

Fig. 1 visualizes the posterior scores and the sampled points based on the five methods. It is observed that the posterior scores estimated using methods (a) and (b) are significantly different from the true posterior scores measured by method (e). This causes the sampled points in methods (a) and (b) to deviate from those sampled based on method (e). On the other hand, the estimated posterior scores and the sampled points in method (c) are relatively similar to those in method (e). The above results therefore suggest that the score mismatch issue is severe under the cases of methods (a) and (b), but is alleviated when method (c) is used.

In order to inspect the potential causes for the differences between the results produced by methods (a), (b), and (c), we incorporate metrics for evaluating the sampling quality and the errors between the scores in an quantitative manner. The sampling quality is evaluated using the precision and recall (Kynkäänniemi et al., 2019) metrics. On the other hand, the estimation errors of the score functions are measured by the expected values of $D_P$ and $D_L$, which are formulated according to the Euclidean distances between the estimated scores and the oracle scores. The expressions of $D_P$ and $D_L$ are represented as the following:

$$
\begin{aligned}
D_P(\tilde{\boldsymbol{x}}, \tilde{\boldsymbol{y}}) &= \|S_P(\tilde{\boldsymbol{x}}, \tilde{\boldsymbol{y}}) - \nabla_{\tilde{\boldsymbol{x}}} \log p_{\sigma,\tau}(\tilde{\boldsymbol{x}}|\tilde{\boldsymbol{y}})\|, \\
D_L(\tilde{\boldsymbol{x}}, \tilde{\boldsymbol{y}}) &= \|S_L(\tilde{\boldsymbol{x}}, \tilde{\boldsymbol{y}}) - \nabla_{\tilde{\boldsymbol{x}}} \log p_{\sigma,\tau}(\tilde{\boldsymbol{y}}|\tilde{\boldsymbol{x}})\|,
\end{aligned}
\tag{7}
$$

where the subscripts $P$ and $L$ denote 'Posterior' and 'Likelihood', while the terms $S_P(\tilde{\boldsymbol{x}}, \tilde{\boldsymbol{y}})$ and $S_L(\tilde{\boldsymbol{x}}, \tilde{\boldsymbol{y}})$ correspond to the estimated posterior score and the likelihood score for a certain pair $(\tilde{\boldsymbol{x}}, \tilde{\boldsymbol{y}})$, respectively.

Table 1 presents $S_P(\tilde{\boldsymbol{x}}, \tilde{\boldsymbol{y}})$ and $S_L(\tilde{\boldsymbol{x}}, \tilde{\boldsymbol{y}})$, the expectations of $D_P$ and $D_L$, and the precision and recall for different methods. It can be seen that the numerical results in Table 1 are consistent with the observations revealed in Fig. 1, since the expectations of $D_P$ are greater in methods (a) and (b) as compared to method (c), and the evaluated recall values in methods (a) and (b) are lower than those in method (c). Furthermore, the results in Table 1 suggest that the possible reasons for the disparity of the posterior scores between methods (a)~(c) are twofold. First, adding the scaling factor $\alpha$ to the likelihood scores increases the expected distance between the estimated score and oracle score, which may exacerbate the score mismatch issue. Second, since the main difference between methods (a)~(c) lies in their score-matching objectives, i.e., the parameters $\theta$ in methods (a) and (b) are optimized using $L_{\text{CE}}$ only while the parameters $\phi_1$ and $\phi_2$ of the score models in method (c) are

optimized through $L_{\text{DSM}}$, the adoption of the score-matching objective may potentially be the key factor to the success of method (c).

The above experimental clues therefore shed light on two essential issues to be further explored and dealt with. First, although employing a classifier trained with $L_{\text{CE}}$ to assist estimating the oracle posterior score is theoretically feasible, this method may potentially lead to considerable discrepancies in practice. This implies that the score mismatch issue stated in Section 2.4 may be the result of the inaccurate likelihood scores produced by a classifier. Second, the comparisons between methods (a), (b), and (c) suggest that score matching may potentially be the solution to the score mismatch issue. Based on these hypotheses, this paper explores an alternative approach to the previous works, called denoising likelihood score matching, which incorporates the score-matching technique when training the classifier to enhance its capability to capture the true likelihood scores. In the next section, we detail the theoretical derivation and the property of our method.

## 4 DENOISING LIKELIHOOD SCORE MATCHING

In this section, we introduce the proposed denoising likelihood score-matching (DLSM) loss, a new training objective that encourages the classifier to capture the true likelihood score. In Section 4.1, we derive this loss function and discuss the intuitions behind it. In Section 4.2, we elaborate on the training procedure of DLSM, and highlight the key property and implications of this training objective.

### 4.1 THE DERIVATION OF THE DENOISING LIKELIHOOD SCORE MATCHING LOSS

As discussed in Section 3, a score model trained with the score-matching objective can potentially be beneficial in producing a better posterior score estimation. In light of this, a classifier may be enhanced if the score-matching process is involved during its training procedure. An intuitive way to accomplish this aim is through minimizing the explicit likelihood score-matching loss $L_{\text{ELSM}}$, which is defined as the following:

$$L_{\text{ELSM}}(\theta) = \mathbb{E}_{p_{\sigma,\tau}(\tilde{\boldsymbol{x}},\tilde{\boldsymbol{y}})} \left[ \frac{1}{2} \|\nabla_{\tilde{\boldsymbol{x}}} \log p(\tilde{\boldsymbol{y}}|\tilde{\boldsymbol{x}};\theta) - \nabla_{\tilde{\boldsymbol{x}}} \log p_{\sigma,\tau}(\tilde{\boldsymbol{y}}|\tilde{\boldsymbol{x}})\|^2 \right]. \tag{8}$$

This loss term, however, involves the calculation of the true likelihood score $\nabla_{\tilde{\boldsymbol{x}}} \log p_{\sigma,\tau}(\tilde{\boldsymbol{y}}|\tilde{\boldsymbol{x}})$, whose computational cost grows with respect to the dataset size. In order to reduce the computational cost, we follow the derivation of DSM as well as Bayes' theorem, and formulate an alternative objective called DLSM loss ($L_{\text{DLSM}}$):

$$L_{\text{DLSM}}(\theta) = \mathbb{E}_{p_{\sigma,\tau}(\tilde{\boldsymbol{x}},\boldsymbol{x},\tilde{\boldsymbol{y}},\boldsymbol{y})} \left[ \frac{1}{2} \|\nabla_{\tilde{\boldsymbol{x}}} \log p(\tilde{\boldsymbol{y}}|\tilde{\boldsymbol{x}};\theta) + \nabla_{\tilde{\boldsymbol{x}}} \log p_{\sigma}(\tilde{\boldsymbol{x}}) - \nabla_{\tilde{\boldsymbol{x}}} \log p_{\sigma}(\tilde{\boldsymbol{x}}|\boldsymbol{x})\|^2 \right]. \tag{9}$$

**Theorem 1.** $L_{\text{DLSM}}(\theta) = L_{\text{ELSM}}(\theta) + C$, where $C$ is a constant with respect to $\theta$.

The proof of this theorem is provided in Appendix A.5. Theorem 1 suggests that optimizing $\theta$ with $L_{\text{ELSM}}(\theta)$ is equivalent to optimizing $\theta$ with $L_{\text{DLSM}}(\theta)$. In contrast to $L_{\text{ELSM}}$, $L_{\text{DLSM}}$ can be approximated in a computationally feasible fashion. This is because $\nabla_{\tilde{\boldsymbol{x}}} \log p_{\sigma}(\tilde{\boldsymbol{x}})$ can be estimated using a score model $s(\tilde{\boldsymbol{x}};\phi)$ trained with $L_{\text{DSM}}$, and $\nabla_{\tilde{\boldsymbol{x}}} \log p_{\sigma}(\tilde{\boldsymbol{x}}|\boldsymbol{x}) = \frac{1}{\sigma^2}(\boldsymbol{x} - \tilde{\boldsymbol{x}})$ as described in Section 2.3. As $\nabla_{\tilde{\boldsymbol{x}}} \log p_{\sigma}(\tilde{\boldsymbol{x}})$ and $\nabla_{\tilde{\boldsymbol{x}}} \log p_{\sigma}(\tilde{\boldsymbol{x}}|\boldsymbol{x})$ can be computed in a tractable manner, the classifier can be updated by minimizing the approximated variant of $L_{\text{DLSM}}$, defined as:

$$L_{\text{DLSM}'}(\theta) = \mathbb{E}_{p_{\sigma,\tau}(\tilde{\boldsymbol{x}},\boldsymbol{x},\tilde{\boldsymbol{y}},\boldsymbol{y})} \left[ \frac{1}{2} \|\nabla_{\tilde{\boldsymbol{x}}} \log p(\tilde{\boldsymbol{y}}|\tilde{\boldsymbol{x}};\theta) + s(\tilde{\boldsymbol{x}};\phi) - \nabla_{\tilde{\boldsymbol{x}}} \log p_{\sigma}(\tilde{\boldsymbol{x}}|\boldsymbol{x})\|^2 \right]. \tag{10}$$

The underlying intuition of $L_{\text{DLSM}}$ and $L_{\text{DLSM}'}$ is to match the likelihood score via matching the posterior score. More specifically, the sum of the first two terms $\nabla_{\tilde{\boldsymbol{x}}} \log p(\tilde{\boldsymbol{y}}|\tilde{\boldsymbol{x}};\theta)$ and $\nabla_{\tilde{\boldsymbol{x}}} \log p_{\sigma}(\tilde{\boldsymbol{x}})$ in $L_{\text{DLSM}}$ should ideally construct the posterior score (i.e., Eq. (5)). By following the posterior score, the perturbed data sample $\tilde{\boldsymbol{x}}$ should move towards the clean sample $\boldsymbol{x}$, since $\nabla_{\tilde{\boldsymbol{x}}} \log p_{\sigma}(\tilde{\boldsymbol{x}}|\boldsymbol{x})$ is equal to the weighted difference $\frac{1}{\sigma^2}(\boldsymbol{x} - \tilde{\boldsymbol{x}})$ (i.e., Eq. (4)). In the next section, we discuss the training procedure with $L_{\text{DLSM}'}$ as well as the property of it.

Figure 2: The training procedure of the proposed methodology.

## 4.2 THE APPROXIMATED DLSM OBJECTIVE IN THE CLASSIFIER TRAINING PROCESS

Following the theoretical derivation in Section 4.1, we next discuss the practical aspects during training, and propose to train the classifier by jointly minimizing the approximated denoising likelihood score-matching loss $L_{\text{DLSM}'}$ and the cross-entropy loss $L_{\text{CE}}$. In practice, the total training objective of the classifier can be written as follows:

$$L_{\text{Total}}(\theta) = L_{\text{DLSM}'}(\theta) + \lambda L_{\text{CE}}(\theta), \tag{11}$$

where $\lambda > 0$ is a balancing coefficient. According to Theorem 1, employing $L_{\text{DLSM}'}$ allows the gradients of the classifier to estimate the likelihood scores. However, since the computation of the term $L_{\text{DLSM}'}$ requires the approximation of $\nabla_{\tilde{\boldsymbol{x}}} \log p_\sigma(\tilde{\boldsymbol{x}})$ from a score model $s(\tilde{\boldsymbol{x}}; \phi)$, the training process that utilizes $L_{\text{DLSM}'}$ alone is undesirable due to its instability on real-world datasets. To reinforce the stability of it, an additional cross-entropy loss $L_{\text{CE}}$ is adopted during the training process of the classifier. The advantage of $L_{\text{CE}}$ is that it leverages the ground truth labels to assist the classifier to learn to match the true likelihood density $p_{\sigma,\tau}(\tilde{\boldsymbol{y}}|\tilde{\boldsymbol{x}})$, which in turn helps the score estimation to be improved. To validate the above advantage, an ablation analysis are offered in Section 5.3 to demonstrate the differences between the classifiers trained with $L_{\text{Total}}$, $L_{\text{CE}}$, and $L_{\text{DLSM}'}$. The analysis reveals that $L_{\text{Total}}$ does provide the best score-matching results while maintaining the stability.

Fig. 2 depicts a two-stage training procedure adopted in this work. In stage 1, a score model $s(\tilde{\boldsymbol{x}}; \phi)$ is updated using $L_{\text{DSM}}$ to match $\nabla_{\tilde{\boldsymbol{x}}} \log p_\sigma(\tilde{\boldsymbol{x}})$ (i.e., Eq. (4)). In stage 2, the weights of the trained score model are fixed, and a classifier $p(\tilde{\boldsymbol{y}}|\tilde{\boldsymbol{x}}; \theta)$ is updated using $L_{\text{Total}}$. After these two training stages, $s(\tilde{\boldsymbol{x}}; \phi)$ and $\nabla_{\tilde{\boldsymbol{x}}} \log p(\tilde{\boldsymbol{y}}|\tilde{\boldsymbol{x}}; \theta)$ can then be added together based on Eq. (6) to perform conditional sampling.

## 5 EXPERIMENTS

In this section, we present experiments to validate the effectiveness of the proposed method. First, we provide the experimental configurations and describe the baseline methods. Next, we evaluate the proposed method quantitatively against the baselines, and demonstrate its improved performance on the Cifar-10 and Cifar-100 datasets. Lastly, we offer an ablation analysis to inspect the impacts of $L_{\text{CE}}$ and $L_{\text{DLSM}'}$ in $L_{\text{Total}}$.

### 5.1 EXPERIMENTAL SETUPS

In this work, we quantitatively compare the base method, scaling method, and our method (i.e., (a), (b), and (d) in Section 3) on the Cifar-10 and Cifar-100 datasets, which contain real-world RGB images with 10 and 100 categories of objects, respectively. For both of the datasets, the image size is $32{\times}32{\times}3$, and the pixel values in an image are first rescaled to $[-1, 1]$ before training. For the sampling algorithm, we adopt the predictor-corrector (PC) sampler described in (Song et al., 2021b) with the sampling steps set to $T$=1,000. The score model architecture is exactly the same as the one used in (Song et al., 2021b), while the architecture of the classifier is based on ResNet (He et al., 2016) with a conditional branch for encoding the information of the standard deviation $\sigma$ (Song et al., 2021b). When training the classifier, $\tau$ is assumed to be sufficiently small, thus the unperturbed labels $\boldsymbol{y}$ are adopted. Furthermore, the balancing coefficient $\lambda$ is set to 1 for the Cifar-10 and Cifar-100 datasets, and 0.125 for the motivational example. For more details about our implementation and additional experimental results, please refer to Appendix A.6 and Appendix A.7.

Table 2: The evaluation results on the Cifar-10 and Cifar-100 datasets. The P / R / D / C metrics with '(CW)' in the last four rows represents the average class-wise metrics described in Section 5.2. The arrow symbols ↑ / ↓ represent that a higher / lower evaluation result correspond to a better performance.

|  |  | Cifar-10 | | | Cifar-100 | | |
| --- | --- | --- | --- | --- | --- | --- | --- |
|  |  | Base | Scaling | Ours | Base | Scaling | Ours |
| FID | ↓ | 4.10 | 12.48 | **2.25** | 4.52 | 12.58 | **3.86** |
| IS | ↑ | 9.08 | 9.37 | **9.90** | 11.53 | 11.59 | **11.62** |
| Precision | ↑ | 0.67 | **0.75** | 0.65 | 0.62 | **0.65** | 0.61 |
| Recall | ↑ | 0.61 | 0.49 | **0.62** | 0.62 | 0.52 | **0.63** |
| Density | ↑ | 1.05 | **1.36** | 0.96 | 0.84 | **0.93** | 0.82 |
| Coverage | ↑ | 0.80 | 0.75 | **0.81** | **0.71** | 0.61 | **0.71** |
| CAS | ↑ | 0.38 | 0.46 | **0.58** | 0.15 | 0.24 | **0.28** |
| (CW) Precision | ↑ | 0.51 | **0.70** | 0.56 | 0.33 | **0.59** | 0.42 |
| (CW) Recall | ↑ | 0.59 | 0.42 | **0.61** | **0.68** | 0.41 | **0.68** |
| (CW) Density | ↑ | 0.63 | **1.23** | 0.76 | 0.30 | **0.78** | 0.43 |
| (CW) Coverage | ↑ | 0.60 | 0.66 | **0.71** | 0.38 | 0.51 | **0.52** |

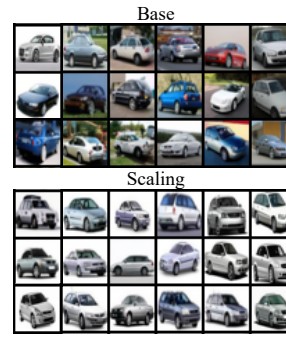

Figure 3: A comparison of the curated samples generated via the base method and the scaling method. As for the uncurated visualized samples, please refer to Appendix A.7.4.

## 5.2 RESULTS ON CIFAR-10 AND CIFAR-100

In this section, we examine the effectiveness of the base method, the scaling method, and our proposed method on the Cifar-10 and Cifar-100 benchmarks with several key evaluation metrics. We adopt the Inception Score (IS) (Barratt & Sharma, 2018) and the Fréchet Inception Distance (FID) (Heusel et al., 2017) as the metrics for evaluating the overall sampling quality by comparing the similarity between the distributions of the generated images and the real images. We also evaluate the methods using the Precision (P), Recall (R) (Kynkäänniemi et al., 2019), Density (D), and Coverage (C) (Naeem et al., 2020) metrics to further examine the fidelity and diversity of the generated images. In addition, we report the Classification Accuracy Score (CAS) (Ravuri & Vinyals, 2019) to measure if the generated samples bear representative class information. Given a dataset containing $m_i$ images for each class $i$, we first conditionally generate the same number of images (i.e., $m_i$) for each class. The FID, IS, P / R / D / C, and CAS metrics are then evaluated based on all the generated $m = \sum_{i=1}^{c} m_i$ images, where $c$ is the total number of classes. In other words, the above metrics are evaluated in an unconditional manner.

Table 2 reports the quantitative results of the above methods. It is observed that the proposed method outperforms the other two methods with substantial margins in terms of FID and IS, indicating that the generated samples bear closer resemblance to the real data. Meanwhile, for the P / R / D / C metrics, the scaling method is superior to the other two methods in terms of the fidelity metrics (i.e., precision and density). However, this method may cause the diversity of the generated images to degrade, as depicted in Fig. 3, resulting in significant performance drops for the diversity metrics (i.e., the recall and the coverage metrics).

Another insight is that the base method achieves relatively better performance on the precision and density metrics in comparison to our method. However, it fails to deliver analogous tendency on the CAS metric. This behavior indicates that the base method may be susceptible to generating false positive samples, since the evaluation of the P / R / D / C metrics does not involve the class information, and thus may fail to consider samples with wrong classes. Such a phenomenon motivates us to further introduce a set of class-wise (CW) metrics, which takes the class information into account by evaluating the P / R / D / C metrics on a per-class basis. Specifically, the class-wise metrics are evaluated separately for each class $i$ with $m_i$ images. The evaluation results of these metrics shown in Fig. 4 reveal that our method outperforms the base method for a majority of classes. Moreover, the results of the average class-wise metrics presented in the last four rows of Table 2 also show that our method yields a better performance as compared to the base method.

Base on these evidences, it can be concluded that the proposed method outperforms both baseline methods in terms of FID and IS, implying that our method does possess a better ability to capture the true data distribution. Additionally, the evaluation results on the CAS and the class-wise metrics suggest that our method does offer a superior ability for a classifier to learn accurate class information as compared to the base method.

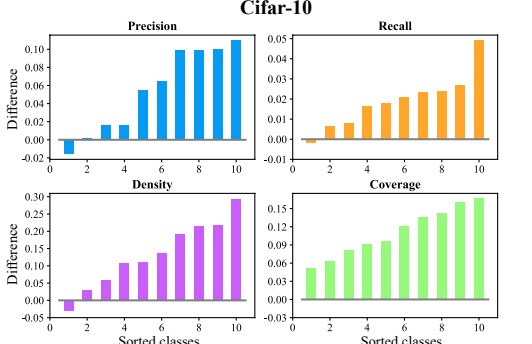
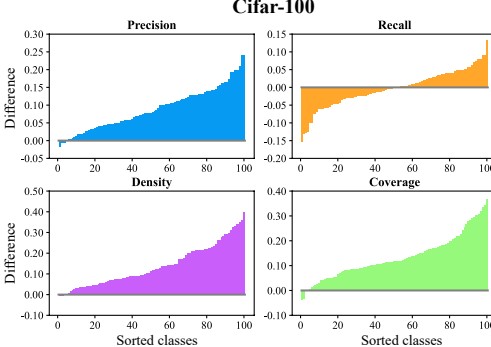

Figure 4: The sorted differences between the proposed method and the base method evaluated on the Cifar-10 and Cifar-100 datasets for the class-wise P / R / D / C metrics. Each colored bar in the plots represents the differences between our method and the base method evaluated using one of the P / R / D / C metrics for a certain class. A positive difference represents that our method outperforms the base method for that class.

## 5.3 ABLATION ANALYSIS

To further investigate the characteristic of $L_{\text{Total}}$, we perform an ablation analysis on the two components $L_{\text{CE}}$ and $L_{\text{DLSM}'}$ in $L_{\text{Total}}$ using the inter-twinning moon dataset, and observe the trends of (a) the score errors $\frac{1}{2}\mathbb{E}_{U(\tilde{\boldsymbol{x}})}[D_L(\tilde{\boldsymbol{x}}, c_1)] + \frac{1}{2}\mathbb{E}_{U(\tilde{\boldsymbol{x}})}[D_L(\tilde{\boldsymbol{x}}, c_2)]$, and (b) the cross entropy $\mathbb{E}_{p_{\sigma,\tau}(\tilde{\boldsymbol{x}},\tilde{\boldsymbol{y}})}[-\log p(\tilde{\boldsymbol{y}}|\tilde{\boldsymbol{x}};\theta)]$ during the training iterations. Metrics (a) and (b) measure the accuracies of the estimated likelihood scores $\nabla_{\tilde{\boldsymbol{x}}} \log p(\tilde{\boldsymbol{y}}|\tilde{\boldsymbol{x}};\theta)$ and the estimated likelihood density $p(\tilde{\boldsymbol{y}}|\tilde{\boldsymbol{x}};\theta)$, respectively. As depicted in Fig 5, an increasing trend is observed for metric (a) when the classifier is trained with $L_{\text{CE}}$ alone. On the contrary, the opposite tendency is observed for those trained with $L_{\text{Total}}$ and $L_{\text{DLSM}'}$. The results thus suggest that matching the likelihood scores implicitly through training a classifier with $L_{\text{CE}}$ alone leads to larger approximation errors. In contrast, $L_{\text{DLSM}'}$ and $L_{\text{Total}}$ explicitly encourage the classifier to capture accurate likelihood scores, since they involve the score-matching

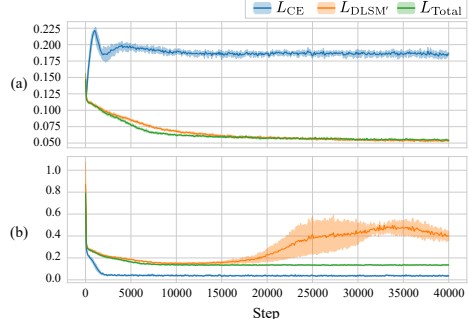

Figure 5: The evaluation curves of (a) the score errors and (b) the cross entropy during the training iterations for $L_{\text{CE}}$, $L_{\text{DLSM}'}$, and $L_{\text{Total}}$. The curves depict the mean and 95% confidence interval of five times of training.

objective. On the other hand, for metric (b), the classifiers trained with $L_{\text{Total}}$ and $L_{\text{CE}}$ yield stable decreasing trend in comparison to that trained with $L_{\text{DLSM}'}$ alone. These results suggest that in addition to minimizing $L_{\text{DLSM}'}$ between $\nabla_{\tilde{\boldsymbol{x}}} \log p_{\sigma,\tau}(\tilde{\boldsymbol{y}}|\tilde{\boldsymbol{x}})$ and $\nabla_{\tilde{\boldsymbol{x}}} \log p(\tilde{\boldsymbol{y}}|\tilde{\boldsymbol{x}};\theta)$, the utilization of $L_{\text{CE}}$ as an auxiliary objective enhances the stability during training. Based on the above observations, the interplay of $L_{\text{CE}}$ and $L_{\text{DLSM}'}$ synergistically achieves the best results in terms of metric (a) while ensuring the stability of the training process. These clues thus validate the adoption of $L_{\text{Total}}$ during the training of the classifier.

## 6 CONCLUSION

In this paper, we highlighted the score mismatch issue in the existing conditional score-based data generation methods, and theoretically derived a new denoising likelihood score-matching (DLSM) loss, which is a training objective for the classifier to match the true likelihood score. We have demonstrated that, by adopting the proposed DLSM loss, the likelihood scores can be better estimated, and the negative impact of the score mismatch issue can be alleviated. Our experimental results validated that the proposed method does offer benefits in producing higher-quality conditional sampling results on both Cifar-10 and Cifar-100 datasets.

ACKNOWLEDGMENTS

This work was supported by the Ministry of Science and Technology (MOST) in Taiwan under grant number MOST 111-2628-E-007-010. The authors acknowledge the financial support from MediaTek Inc., Taiwan. The authors would also like to acknowledge the donation of the GPUs from NVIDIA Corporation and NVIDIA AI Technology Center (NVAITC) used in this research work. The authors thank National Center for High-Performance Computing (NCHC) for providing computational and storage resources. Finally, the authors would also like to thank the time and effort of the annoymous reviewers for reviewing this paper.

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

## A  APPENDIX

In this Appendix, we first provide the definitions for the symbols used in the main manuscript and the Appendix in Section A.1. Next, we derive the theoretical proof for the closed form of $\nabla_{\tilde{x}} \log p_\sigma(\tilde{x})$ in Section A.2. Then, we offer a number of discussions on the scaling technique in Section A.3, and different conditional score-based data generation methods in Section A.4. Subsequently, we offer the theoretical proof for Theorem 1 in Section A.5. Finally, in Sections A.6 and A.7, we provide the detailed experimental setups and additional experimental results.

### A.1  THE LIST OF SYMBOLS USED IN THE PAPER

In this section, we provide the list of symbols used throughout the main manuscript and the Appendix. These symbols are summarized in Tables A1 and A2, containing their notations as well as their detailed descriptions.

| Symbol | Description |
|---|---|
| $\langle \boldsymbol{u}, \boldsymbol{v} \rangle = \boldsymbol{u}^T \boldsymbol{v} = \sum_i \boldsymbol{u}_i \boldsymbol{v}_i$ | dot (inner) product between two vectors. |
| $\|\boldsymbol{u}\| = \sqrt{\langle \boldsymbol{u}, \boldsymbol{u} \rangle}$ | Euclidean norm of vector. |
| $\boldsymbol{I}_{d \times d}$ | identity matrix with dimension $d \times d$. |
| $\epsilon$ | the step size used in Langevin Diffusion. |
| $t \in \{0, ..., T\}$ | the discrete timestep used in Langevin Diffusion. |
| $\boldsymbol{x}, \boldsymbol{x}_t \in \mathbb{R}^d$ | data sample with dimension $d$, where $\boldsymbol{x}_0$ is sampled from $\mathcal{N}(\boldsymbol{0}, \boldsymbol{I}_{d \times d})$. |
| $\tilde{\boldsymbol{x}} \in \mathbb{R}^d$ | perturbed data sample used in Denoising Score Matching. |
| $\boldsymbol{y} \in \mathbb{R}^c$ | conditional class label with dimension $c$. |
| $\tilde{\boldsymbol{y}} \in \mathbb{R}^c$ | perturbed class label used in Denoising Score Matching. |
| $\boldsymbol{z} : \mathbb{R}^d$ | the noise vector used in Langevin Diffusion. (sampled from $\mathcal{N}(\boldsymbol{0}, \boldsymbol{I}_{d \times d})$) |
| $\boldsymbol{x}^{(i)} \in \mathbb{R}^d$ | the $i$-th example (input) from a dataset $\mathbb{D}$. |
| $\boldsymbol{y}^{(i)} \in \{0,1\}^c$ | the class label associated with $\boldsymbol{x}^{(i)}$ for supervised learning (one-hot encoded). |
| $\mathbb{D} = \{(\boldsymbol{x}^{(1)}, \boldsymbol{y}^{(1)}), ..., (\boldsymbol{x}^{(m)}, \boldsymbol{y}^{(m)})\}$ | Training set: $m$ data-label pairs from $p_{\text{data}}$. (i.i.d. samples from $p_{\text{data}}$) |
| $p_{\text{data}}(\boldsymbol{x})$ | unknown true probability density function (pdf) of data samples. |
| $p_0(\boldsymbol{x}) = \frac{1}{m} \sum_{i=1}^m \delta(\|\boldsymbol{x} - \boldsymbol{x}^{(i)}\|)$ | empirical pdf of data samples associated with the training set $\mathbb{D}$. |
| $p_\sigma(\tilde{\boldsymbol{x}}|\boldsymbol{x}) = \frac{1}{(2\pi)^{d/2}\sigma^d} e^{\frac{-1}{2\sigma^2}\|\tilde{\boldsymbol{x}} - \boldsymbol{x}\|^2}$ | smoothing kernel or noise model: sampled from isotropic Gaussian $\mathcal{N}(\boldsymbol{x}, \sigma^2 \boldsymbol{I}_{d \times d})$ with mean $\boldsymbol{x}$ and variance $\sigma^2$. |
| $p_\sigma(\tilde{\boldsymbol{x}}, \boldsymbol{x}) = p_\sigma(\tilde{\boldsymbol{x}}|\boldsymbol{x})p_0(\boldsymbol{x})$ | joint pdf of perturbed data and clean data samples. |
| $p_\sigma(\tilde{\boldsymbol{x}}) = \frac{1}{m} \sum_{i=1}^m p_\sigma(\tilde{\boldsymbol{x}}|\boldsymbol{x}^{(i)})$ | Parzen density estimate based on $\mathbb{D}$, obtainable by marginalizing $p_\sigma(\tilde{\boldsymbol{x}}, \boldsymbol{x})$. |
| $p_{\text{data}}(\boldsymbol{y})$ | unknown true pdf of data labels. |
| $p_0(\boldsymbol{y}) = \frac{1}{m} \sum_{i=1}^m \delta(\|\boldsymbol{y} - \boldsymbol{y}^{(i)}\|)$ | empirical pdf of data labels associated with the training set $\mathbb{D}$. |
| $p_\tau(\tilde{\boldsymbol{y}}|\boldsymbol{y}) = \frac{1}{(2\pi)^{c/2}\tau^c} e^{\frac{-1}{2\tau^2}\|\tilde{\boldsymbol{y}} - \boldsymbol{y}\|^2}$ | smoothing kernel or noise model: sampled from isotropic Gaussian $\mathcal{N}(\boldsymbol{y}, \tau^2 \boldsymbol{I}_{c \times c})$ with mean $\boldsymbol{y}$ and variance $\tau^2$. |
| $p_\tau(\tilde{\boldsymbol{y}}, \boldsymbol{y}) = p_\tau(\tilde{\boldsymbol{y}}|\boldsymbol{y})p_0(\boldsymbol{y})$ | joint pdf of perturbed labels and clean labels. |
| $p_\tau(\tilde{\boldsymbol{y}}) = \frac{1}{m} \sum_{i=1}^m p_\tau(\tilde{\boldsymbol{y}}|\boldsymbol{y}^{(i)})$ | Parzen density estimate based on $\mathbb{D}$, obtainable by marginalizing $p_\tau(\tilde{\boldsymbol{y}}, \boldsymbol{y})$. |
| $p_{0,0}(\boldsymbol{x}, \boldsymbol{y})$ | defined as $\frac{1}{m} \sum_{i=1}^m \delta(\|\boldsymbol{x} - \boldsymbol{x}^{(i)}\|)\delta(\|\boldsymbol{y} - \boldsymbol{y}^{(i)}\|)$. |
| $p_{\sigma,0}(\tilde{\boldsymbol{x}}, \boldsymbol{x}, \boldsymbol{y})$ | equals to $p_\sigma(\tilde{\boldsymbol{x}}|\boldsymbol{x})p_{0,0}(\boldsymbol{x}, \boldsymbol{y})$. |
| $p_{0,\tau}(\boldsymbol{x}, \tilde{\boldsymbol{y}}, \boldsymbol{y})$ | equals to $p_\tau(\tilde{\boldsymbol{y}}|\boldsymbol{y})p_{0,0}(\boldsymbol{x}, \boldsymbol{y})$. |
| $p_{\sigma,\tau}(\tilde{\boldsymbol{x}}, \boldsymbol{x}, \tilde{\boldsymbol{y}}, \boldsymbol{y})$ | defined as $p_\sigma(\tilde{\boldsymbol{x}}|\boldsymbol{x})p_\tau(\tilde{\boldsymbol{y}}|\boldsymbol{y})p_{0,0}(\boldsymbol{x}, \boldsymbol{y})$. |
| $s(\tilde{\boldsymbol{x}}; \phi)$ | Score Model: approximated score function with parameters $\phi$. |
| $p(\tilde{\boldsymbol{y}}|\tilde{\boldsymbol{x}}; \theta)$ | Classifier: approximated probability function parameterized by $\theta$. |

Table A1: The list of symbols used in this paper.

| $\alpha$ | the scaling factor. |
|---|---|
| $\nabla_{\tilde{x}} \log p(\tilde{x}|y)$ | posterior score. |
| $\nabla_{\tilde{x}} \log p(y|\tilde{x})$ | likelihood score. |
| $\nabla_{\tilde{x}} \log p(\tilde{x})$ | prior score. |
| $L_{\text{ESM}}(\phi)$ | Explicit Score Matching (ESM) loss for optimizing $s(\tilde{x}; \phi)$. |
| $L_{\text{DSM}}(\phi)$ | Denoising Score Matching loss (DSM) for optimizing $s(\tilde{x}; \phi)$. |
| $L_{\text{ELSM}}(\theta)$ | Explicit Likelihood Score Matching (ELSM) loss for optimizing $p(y|\tilde{x}; \theta)$. |
| $L_{\text{DLSM}}(\theta)$ | Denoising Likelihood Score Matching (DLSM) loss for optimizing $p(y|\tilde{x}; \theta)$. |
| $L_{\text{DLSM}'}(\theta)$ | approximated Denoising Likelihood Score Matching loss for optimizing $p(y|\tilde{x}; \theta)$. |
| $L_{\text{CE}}(\theta)$ | Cross Entropy (CE) loss for optimizing $p(y|\tilde{x}; \theta)$. |
| $L_{\text{Total}}(\theta)$ | full loss for optimizing $p(y|\tilde{x}; \theta)$. |

Table A2: The list of symbols used in this paper.

### A.2   A PROOF OF THE CLOSED FORM FOR $\nabla_{\tilde{x}} \log p_\sigma(\tilde{x})$

In Sections 2.2 and 3, we utilized the closed form formula to calculate the true scores $\nabla_{\tilde{x}} \log p_\sigma(\tilde{x})$. To show the equivalence holds for Eq. (2), we provide a formal derivation as shown in the following proposition.

**Proposition 1.** *Given* $p_\sigma(\tilde{x}) = \frac{1}{m} \sum_{i=1}^{m} p_\sigma(\tilde{x}|x^{(i)})$, $p_\sigma(\tilde{x}|x) = \frac{1}{(2\pi)^{d/2}\sigma^d} e^{\frac{-1}{2\sigma^2}\|\tilde{x}-x\|^2}$, *and* $\sigma > 0$, *the closed form of* $\nabla_{\tilde{x}} \log p_\sigma(\tilde{x})$ *can be represented as follows:*

$$\nabla_{\tilde{x}} \log p_\sigma(\tilde{x}) = \frac{\sum_{i=1}^{m} \frac{1}{\sigma^2}(x^{(i)} - \tilde{x})p_\sigma(\tilde{x}|x^{(i)})}{\sum_{i=1}^{m} p_\sigma(\tilde{x}|x^{(i)})}.$$

*Proof.*

$$\nabla_{\tilde{x}} \log p_\sigma(\tilde{x}) = \nabla_{\tilde{x}} \log \frac{1}{m} \sum_{i=1}^{m} p_\sigma(\tilde{x}|x^{(i)})$$

$$= \nabla_{\tilde{x}} \log \sum_{i=1}^{m} p_\sigma(\tilde{x}|x^{(i)}) + \underbrace{\nabla_{\tilde{x}} \log \frac{1}{m}}_{=0}$$

$$= \frac{\nabla_{\tilde{x}} \sum_{i=1}^{m} p_\sigma(\tilde{x}|x^{(i)})}{\sum_{i=1}^{m} p_\sigma(\tilde{x}|x^{(i)})}$$

$$= \frac{\sum_{i=1}^{m} \nabla_{\tilde{x}} p_\sigma(\tilde{x}|x^{(i)})}{\sum_{i=1}^{m} p_\sigma(\tilde{x}|x^{(i)})}$$

$$= \frac{\sum_{i=1}^{m} \nabla_{\tilde{x}} \left[ \frac{1}{(2\pi)^{d/2}\sigma^d} e^{\frac{-1}{2\sigma^2}\|\tilde{x}-x^{(i)}\|^2} \right]}{\sum_{i=1}^{m} p_\sigma(\tilde{x}|x^{(i)})}$$

$$= \frac{\sum_{i=1}^{m} \left[ \frac{(x^{(i)}-\tilde{x})/\sigma^2}{(2\pi)^{d/2}\sigma^d} e^{\frac{-1}{2\sigma^2}\|\tilde{x}-x^{(i)}\|^2} \right]}{\sum_{i=1}^{m} p_\sigma(\tilde{x}|x^{(i)})}$$

$$= \frac{\sum_{i=1}^{m} \frac{1}{\sigma^2}(x^{(i)} - \tilde{x})p_\sigma(\tilde{x}|x^{(i)})}{\sum_{i=1}^{m} p_\sigma(\tilde{x}|x^{(i)})}$$

$\square$

### A.3   A DISCUSSION ON THE IMPACTS OF THE SCALING TECHNIQUE

In this section, we first provide a proof to demonstrate the reason behind the non-equivalence between the scaling technique and the re-normalization of a likelihood density. Then, we present the

derivation of the posterior scores based on a re-normalized likelihood density. Finally, we adopt a low-dimensional example to explain the effect of re-normalization on a posterior density.

The authors in (Dhariwal & Nichol, 2021) leveraged a scaling factor $\alpha > 0$ to adjust the likelihood score $\nabla_{\boldsymbol{x}} \log p(\boldsymbol{y}|\boldsymbol{x})$. To demonstrate the effect of such a scaling factor $\alpha$, the authors claimed that there always exists a normalizing constant $Z$ such that $\alpha \nabla_{\boldsymbol{x}} \log p(\boldsymbol{y}|\boldsymbol{x}) = \nabla_{\boldsymbol{x}} \log \frac{1}{Z} p(\boldsymbol{y}|\boldsymbol{x})^\alpha$, where $\frac{1}{Z} p(\boldsymbol{y}|\boldsymbol{x})^\alpha$ is a re-normalized likelihood density. However, such an equality only holds for specific conditions, and is not applicable to general cases, as explained in **Proposition 2**.

**Proposition 2.** *Given a constant $\alpha > 0$, and a probability density function (pdf) $p(\boldsymbol{y}|\boldsymbol{x})$ that is everywhere non-zero and differentiable, there does not exist a constant $Z$ such that $\frac{1}{Z} p(\boldsymbol{y}|\boldsymbol{x})^\alpha$ is a pdf for all $\boldsymbol{x}$ in general.*

*Proof.* The proposition can be proved by contradiction. Assume that such a constant $Z$ exists and consider the following case:

$$\alpha = 2,$$

$$x, y \in \mathbb{R},$$

$$f(y; \mu, \sigma^2) = (2\pi\sigma^2)^{\frac{-1}{2}} e^{\frac{-(y-\mu)^2}{2\sigma^2}},$$

$$p(y|x) = \frac{1}{2(x^2+1)} \left( (x+1)^2 f(y; 0, 1) + (x-1)^2 f(y; 0, \frac{1}{2}) \right).$$

Note that $p(y|x)$ is a pdf and is everywhere non-zero and differentiable.

Since a pdf sums to 1 (i.e., $\int_y \frac{1}{Z} p(y|x)^\alpha dy = 1, \forall x$), by utilizing the Gaussian integral and the above assumption, the following derivation holds:

$$1 = \int_y \frac{1}{Z} p(y|(-1))^\alpha dy = \int_y \frac{1}{Z} p(y|1)^\alpha dy$$

$$\Leftrightarrow Z = \int_y p(y|(-1))^2 dy = \int_y p(y|1)^2 dy$$

$$\Leftrightarrow Z = \int_y f(y; 0, \frac{1}{2})^2 dy = \int_y f(y; 0, 1)^2 dy$$

$$\Leftrightarrow Z = \int_y \left( (\pi)^{\frac{-1}{2}} e^{-y^2} \right)^2 dy = \int_y \left( (2\pi)^{\frac{-1}{2}} e^{\frac{-y^2}{2}} \right)^2 dy$$

$$\Leftrightarrow Z = \frac{1}{\pi} \int_y \left( e^{-y^2} \right)^2 dy = \frac{1}{2\pi} \int_y \left( e^{\frac{-y^2}{2}} \right)^2 dy$$

$$\Leftrightarrow Z = \frac{1}{\pi} \int_y e^{-2y^2} dy = \frac{1}{2\pi} \int_y e^{-y^2} dy$$

$$\Leftrightarrow Z = \frac{1}{\sqrt{2}\pi} \int_u e^{-u^2} du = \frac{1}{2\pi} \int_y e^{-y^2} dy$$

$$\Leftrightarrow Z = \frac{1}{\sqrt{2}\pi} \sqrt{\pi} = \frac{1}{2\pi} \sqrt{\pi}$$

$$\Leftrightarrow Z = \frac{1}{\sqrt{2\pi}} = \frac{1}{2\sqrt{\pi}}$$

The final equation leads to a contradiction for the value of $Z$, which completes the proof of **Proposition 2**. Therefore, we concluded that there does not exist a constant $Z$ such that $\frac{1}{Z} p(\boldsymbol{y}|\boldsymbol{x})^\alpha$ is a pdf for all $\boldsymbol{x}$ in general. $\square$

To the best of our knowledge, the scaling technique still lacks a theoretical justification. However, if $Z$ is allowed to be a function of $\boldsymbol{x}$ rather than a constant, re-normalization of the likelihood density becomes feasible. In other words, setting $Z(\boldsymbol{x}) = \int_{\boldsymbol{y}} p(\boldsymbol{y}|\boldsymbol{x})^\alpha d\boldsymbol{y}$ enables $\int_{\boldsymbol{y}} \frac{1}{Z(\boldsymbol{x})} p(\boldsymbol{y}|\boldsymbol{x})^\alpha d\boldsymbol{y} = 1$,

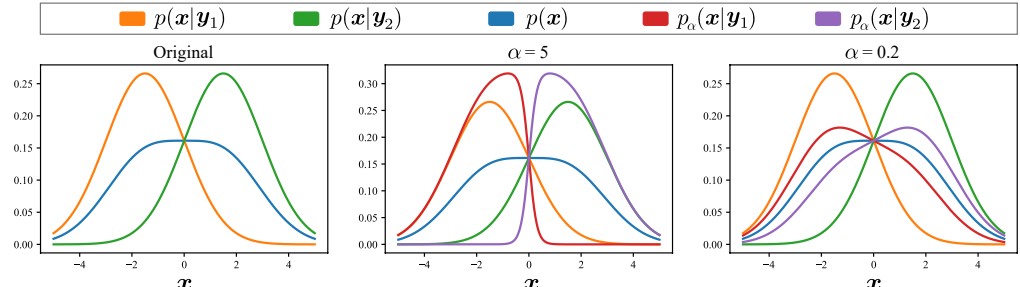

Figure A1: The original and the renormalized densities under different choices of $\alpha$.

where $\frac{1}{Z(\boldsymbol{x})}p(\boldsymbol{y}|\boldsymbol{x})^\alpha \triangleq p_\alpha(\boldsymbol{y}|\boldsymbol{x})$ is a re-normalized likelihood density. In this case, the score of the re-normalized likelihood density can be written as,

$$
\begin{aligned}
\nabla_{\boldsymbol{x}} \log p_\alpha(\boldsymbol{y}|\boldsymbol{x}) &= \nabla_{\boldsymbol{x}} \log \left( \frac{1}{Z(\boldsymbol{x})} p(\boldsymbol{y}|\boldsymbol{x})^\alpha \right) \\
&= \nabla_{\boldsymbol{x}} \Big( \log p(\boldsymbol{y}|\boldsymbol{x})^\alpha - \log Z(\boldsymbol{x}) \Big) \\
&= \nabla_{\boldsymbol{x}} \log p(\boldsymbol{y}|\boldsymbol{x})^\alpha - \nabla_{\boldsymbol{x}} \log Z(\boldsymbol{x}) \\
&= \alpha \nabla_{\boldsymbol{x}} \log p(\boldsymbol{y}|\boldsymbol{x}) - \nabla_{\boldsymbol{x}} \log Z(\boldsymbol{x}).
\end{aligned}
\tag{A1}
$$

Given the joint density $p_\alpha(\boldsymbol{x}, \boldsymbol{y}) \triangleq p_\alpha(\boldsymbol{y}|\boldsymbol{x})p(\boldsymbol{x})$, the posterior of the re-normalized density can be derived by Bayes' Theorem, i.e., $p_\alpha(\boldsymbol{x}|\boldsymbol{y}) = p_\alpha(\boldsymbol{y}|\boldsymbol{x})p(\boldsymbol{x})/\int_{\boldsymbol{x}} p_\alpha(\boldsymbol{x}, \boldsymbol{y})d\boldsymbol{x}$. Based on Eq. (5), the posterior score of the re-normalized density can be written as:

$$
\begin{aligned}
\nabla_{\boldsymbol{x}} \log p_\alpha(\boldsymbol{x}|\boldsymbol{y}) &= \nabla_{\boldsymbol{x}} \log p_\alpha(\boldsymbol{y}|\boldsymbol{x}) + \nabla_{\boldsymbol{x}} \log p(\boldsymbol{x}) \\
&= \alpha \nabla_{\boldsymbol{x}} \log p(\boldsymbol{y}|\boldsymbol{x}) - \nabla_{\boldsymbol{x}} \log Z(\boldsymbol{x}) + \nabla_{\boldsymbol{x}} \log p(\boldsymbol{x}).
\end{aligned}
\tag{A2}
$$

Through the above equation, conditional sampling based on the re-normalized likelihood density becomes feasible.

To further investigate the effect of re-normalization on $p_\alpha(\boldsymbol{x}|\boldsymbol{y})$, we provide an illustrative example in Fig. A1. In this example, the condition $\boldsymbol{y}$ is assumed to be sampled from $\{\boldsymbol{y}_1, \boldsymbol{y}_2\}$ with equal probability $1/2$. The original posterior densities $p(\boldsymbol{x}|\boldsymbol{y}_1)$ and $p(\boldsymbol{x}|\boldsymbol{y}_2)$ are normal distributions with variance $2.25$ centered at $\boldsymbol{x} = -1.5$ and $\boldsymbol{x} = 1.5$, which are plotted in orange and green, respectively. On the other hand, the prior density $p(\boldsymbol{x}) \triangleq \frac{1}{2}p(\boldsymbol{x}|\boldsymbol{y}_1) + \frac{1}{2}p(\boldsymbol{x}|\boldsymbol{y}_2)$ is plotted in blue, and the likelihood density $p(\boldsymbol{y}|\boldsymbol{x}) = p(\boldsymbol{x}|\boldsymbol{y})p(\boldsymbol{y})/p(\boldsymbol{x})$ is defined according to Bayes' theorem.

Subsequently, the re-normalized posterior densities $p_\alpha(\boldsymbol{x}|\boldsymbol{y}_1)$ and $p_\alpha(\boldsymbol{x}|\boldsymbol{y}_2)$ can be constructed based on the equation $\frac{1}{Z(\boldsymbol{x})}p(\boldsymbol{y}_1|\boldsymbol{x})^\alpha$, $\frac{1}{Z(\boldsymbol{x})}p(\boldsymbol{y}_2|\boldsymbol{x})^\alpha$, and Bayes' theorem. In Fig. A1, it is observed that under the case that $\alpha = 5$, the re-normalized posterior densities are more concentrated. Conversely, when $\alpha = 0.2$, the re-normalized posterior densities are much more dispersed. These results therefore suggest that the re-normalized posterior density can be adjusted by controlling the value of $\alpha$.

### A.4 CONDITIONAL SCORE-BASED DATA GENERATION METHODS

As described in Section 3, methods (a), (b), and (d) utilize a classifier $p(\tilde{\boldsymbol{y}}|\tilde{\boldsymbol{x}}; \theta)$ and a score model $s(\tilde{\boldsymbol{x}}; \phi)$ to estimate the posterior scores. In contrast, method (c) separately trains the posterior score models $s(\tilde{\boldsymbol{x}}; \phi_1)$ and $s(\tilde{\boldsymbol{x}}; \phi_2)$ for different class conditions. Although method (c) achieves better evaluation results, it requires training score models for each possible $\tilde{\boldsymbol{y}}$, which is usually considered impractical in real-world applications. A potential solution to this problem is to train a score model $s(\tilde{\boldsymbol{x}}, \tilde{\boldsymbol{y}}; \phi)$ that can estimate the scores for the joint probability of $\tilde{\boldsymbol{x}}$ and $\tilde{\boldsymbol{y}}$ (denoted as 'Method (c*)' in Fig. A2). This method, however, is inflexible in comparison to methods (a), (b), and (d), since any change on the condition $\tilde{\boldsymbol{y}}$ could lead to re-training the joint score model $s(\tilde{\boldsymbol{x}}, \tilde{\boldsymbol{y}}; \phi)$. Therefore, in this work, we only compare the performance of methods (a), (b), and (d), and consider method (c) to be another orthogonal research direction.

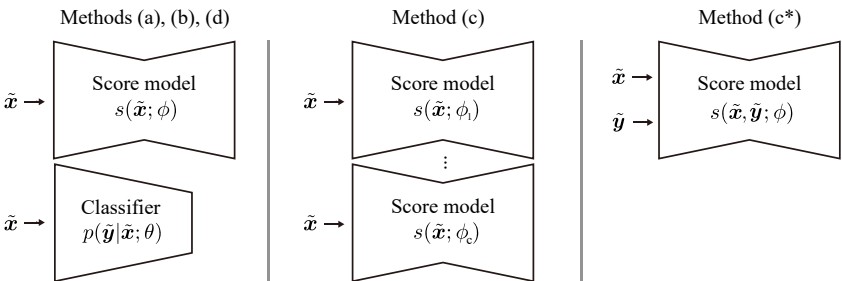

Figure A2: An illustration for different conditional score-based data generation methods in Section A.4.

## A.5 A PROOF FOR DENOISING LIKELIHOOD SCORE MATCHING

In Section 4, we stated the equivalence between $L_{\text{ELSM}}(\theta)$ and $L_{\text{DLSM}}(\theta)$ when optimizing $\theta$. To demonstrate how such an equivalence stands, we provide the theoretical proof as follows.

**Theorem 1.** $L_{\text{DLSM}}(\theta) = L_{\text{ELSM}}(\theta) + C$, where $C$ is a constant with respect to $\theta$.

*Proof.* $L_{\text{ELSM}}(\theta)$ and $L_{\text{DLSM}}(\theta)$ are defined as follows:

$$L_{\text{ELSM}}(\theta) = \mathbb{E}_{p_{\sigma,\tau}(\tilde{\boldsymbol{x}},\tilde{\boldsymbol{y}})} \left[ \frac{1}{2} \| \nabla_{\tilde{\boldsymbol{x}}} \log p(\tilde{\boldsymbol{y}}|\tilde{\boldsymbol{x}};\theta) - \nabla_{\tilde{\boldsymbol{x}}} \log p_{\sigma,\tau}(\tilde{\boldsymbol{y}}|\tilde{\boldsymbol{x}}) \|^2 \right].$$

$$L_{\text{DLSM}}(\theta) = \mathbb{E}_{p_{\sigma,\tau}(\tilde{\boldsymbol{x}},\boldsymbol{x},\tilde{\boldsymbol{y}},\boldsymbol{y})} \left[ \frac{1}{2} \| \nabla_{\tilde{\boldsymbol{x}}} \log p(\tilde{\boldsymbol{y}}|\tilde{\boldsymbol{x}};\theta) + \nabla_{\tilde{\boldsymbol{x}}} \log p_\sigma(\tilde{\boldsymbol{x}}) - \nabla_{\tilde{\boldsymbol{x}}} \log p_\sigma(\tilde{\boldsymbol{x}}|\boldsymbol{x}) \|^2 \right].$$

The proof expands $L_{\text{ELSM}}(\theta)$ and $L_{\text{DLSM}}(\theta)$ individually, leading to constant difference.

$$L_{\text{ELSM}}(\theta) = \mathbb{E}_{p_{\sigma,\tau}(\tilde{\boldsymbol{x}},\tilde{\boldsymbol{y}})} \left[ \frac{1}{2} \| \nabla_{\tilde{\boldsymbol{x}}} \log p(\tilde{\boldsymbol{y}}|\tilde{\boldsymbol{x}};\theta) - \nabla_{\tilde{\boldsymbol{x}}} \log p_{\sigma,\tau}(\tilde{\boldsymbol{y}}|\tilde{\boldsymbol{x}}) \|^2 \right]$$

$$= \mathbb{E}_{p_{\sigma,\tau}(\tilde{\boldsymbol{x}},\tilde{\boldsymbol{y}})} \left[ \frac{1}{2} \| \nabla_{\tilde{\boldsymbol{x}}} \log p(\tilde{\boldsymbol{y}}|\tilde{\boldsymbol{x}};\theta) \|^2 \right] - F(\theta) + C_1, \text{ where}$$

$$C_1 = \mathbb{E}_{p_{\sigma,\tau}(\tilde{\boldsymbol{x}},\tilde{\boldsymbol{y}})} \left[ \frac{1}{2} \| \nabla_{\tilde{\boldsymbol{x}}} \log p_{\sigma,\tau}(\tilde{\boldsymbol{y}}|\tilde{\boldsymbol{x}}) \|^2 \right] \text{ is a constant, and}$$

$$F(\theta) = \mathbb{E}_{p_{\sigma,\tau}(\tilde{\boldsymbol{x}},\tilde{\boldsymbol{y}})} \left[ \langle \nabla_{\tilde{\boldsymbol{x}}} \log p(\tilde{\boldsymbol{y}}|\tilde{\boldsymbol{x}};\theta), \nabla_{\tilde{\boldsymbol{x}}} \log p_{\sigma,\tau}(\tilde{\boldsymbol{y}}|\tilde{\boldsymbol{x}}) \rangle \right]$$

$$= \mathbb{E}_{p_{\sigma,\tau}(\tilde{\boldsymbol{x}},\tilde{\boldsymbol{y}})} \left[ \langle \nabla_{\tilde{\boldsymbol{x}}} \log p(\tilde{\boldsymbol{y}}|\tilde{\boldsymbol{x}};\theta), \nabla_{\tilde{\boldsymbol{x}}} \log p_{\sigma,\tau}(\tilde{\boldsymbol{x}}|\tilde{\boldsymbol{y}}) - \nabla_{\tilde{\boldsymbol{x}}} \log p_\sigma(\tilde{\boldsymbol{x}}) \rangle \right]$$

$$= G(\theta) - \mathbb{E}_{p_{\sigma,\tau}(\tilde{\boldsymbol{x}},\tilde{\boldsymbol{y}})} \left[ \langle \nabla_{\tilde{\boldsymbol{x}}} \log p(\tilde{\boldsymbol{y}}|\tilde{\boldsymbol{x}};\theta), \nabla_{\tilde{\boldsymbol{x}}} \log p_\sigma(\tilde{\boldsymbol{x}}) \rangle \right], \text{ where}$$

$$G(\theta) = \mathbb{E}_{p_{\sigma,\tau}(\tilde{\boldsymbol{x}},\tilde{\boldsymbol{y}})} \left[ \langle \nabla_{\tilde{\boldsymbol{x}}} \log p(\tilde{\boldsymbol{y}}|\tilde{\boldsymbol{x}};\theta), \nabla_{\tilde{\boldsymbol{x}}} \log p_{\sigma,\tau}(\tilde{\boldsymbol{x}}|\tilde{\boldsymbol{y}}) \rangle \right]$$

$$= \int_{\tilde{\boldsymbol{x}}} \int_{\tilde{\boldsymbol{y}}} p_{\sigma,\tau}(\tilde{\boldsymbol{x}},\tilde{\boldsymbol{y}}) \langle \nabla_{\tilde{\boldsymbol{x}}} \log p(\tilde{\boldsymbol{y}}|\tilde{\boldsymbol{x}};\theta), \nabla_{\tilde{\boldsymbol{x}}} \log p_{\sigma,\tau}(\tilde{\boldsymbol{x}}|\tilde{\boldsymbol{y}}) \rangle d\tilde{\boldsymbol{y}} d\tilde{\boldsymbol{x}}$$

$$= \int_{\tilde{\boldsymbol{x}}} \int_{\tilde{\boldsymbol{y}}} p_\tau(\tilde{\boldsymbol{y}}) p_{\sigma,\tau}(\tilde{\boldsymbol{x}}|\tilde{\boldsymbol{y}}) \langle \nabla_{\tilde{\boldsymbol{x}}} \log p(\tilde{\boldsymbol{y}}|\tilde{\boldsymbol{x}};\theta), \frac{\nabla_{\tilde{\boldsymbol{x}}} p_{\sigma,\tau}(\tilde{\boldsymbol{x}}|\tilde{\boldsymbol{y}})}{p_{\sigma,\tau}(\tilde{\boldsymbol{x}}|\tilde{\boldsymbol{y}})} \rangle d\tilde{\boldsymbol{y}} d\tilde{\boldsymbol{x}}$$

$$= \int_{\tilde{\boldsymbol{x}}} \int_{\tilde{\boldsymbol{y}}} p_\tau(\tilde{\boldsymbol{y}}) \langle \nabla_{\tilde{\boldsymbol{x}}} \log p(\tilde{\boldsymbol{y}}|\tilde{\boldsymbol{x}};\theta), \nabla_{\tilde{\boldsymbol{x}}} p_{\sigma,\tau}(\tilde{\boldsymbol{x}}|\tilde{\boldsymbol{y}}) \rangle d\tilde{\boldsymbol{y}} d\tilde{\boldsymbol{x}}$$

$$= \int_{\tilde{\boldsymbol{x}}} \int_{\tilde{\boldsymbol{y}}} p_\tau(\tilde{\boldsymbol{y}}) \langle \nabla_{\tilde{\boldsymbol{x}}} \log p(\tilde{\boldsymbol{y}}|\tilde{\boldsymbol{x}};\theta), \nabla_{\tilde{\boldsymbol{x}}} \int_{\boldsymbol{x}} p_{0,\tau}(\boldsymbol{x}|\tilde{\boldsymbol{y}}) p_{\sigma,\tau}(\tilde{\boldsymbol{x}}|\boldsymbol{x},\tilde{\boldsymbol{y}}) d\boldsymbol{x} \rangle d\tilde{\boldsymbol{y}} d\tilde{\boldsymbol{x}}$$

$$= \int_{\tilde{\boldsymbol{x}}} \int_{\tilde{\boldsymbol{y}}} p_\tau(\tilde{\boldsymbol{y}}) \langle \nabla_{\tilde{\boldsymbol{x}}} \log p(\tilde{\boldsymbol{y}}|\tilde{\boldsymbol{x}};\theta), \nabla_{\tilde{\boldsymbol{x}}} \int_{\boldsymbol{x}} \int_{\boldsymbol{y}} p_{0,\tau}(\boldsymbol{x}|\tilde{\boldsymbol{y}}) p_{0,\tau}(\boldsymbol{y}|\boldsymbol{x},\tilde{\boldsymbol{y}}) p_{\sigma,\tau}(\tilde{\boldsymbol{x}}|\boldsymbol{x},\tilde{\boldsymbol{y}},\boldsymbol{y}) d\boldsymbol{y} d\boldsymbol{x} \rangle d\tilde{\boldsymbol{y}} d\tilde{\boldsymbol{x}}$$

$$= \int_{\tilde{\boldsymbol{x}}} \int_{\tilde{\boldsymbol{y}}} p_\tau(\tilde{\boldsymbol{y}}) \langle \nabla_{\tilde{\boldsymbol{x}}} \log p(\tilde{\boldsymbol{y}}|\tilde{\boldsymbol{x}};\theta), \int_{\boldsymbol{x}} \int_{\boldsymbol{y}} p_{0,\tau}(\boldsymbol{x}|\tilde{\boldsymbol{y}}) p_{0,\tau}(\boldsymbol{y}|\boldsymbol{x},\tilde{\boldsymbol{y}}) \nabla_{\tilde{\boldsymbol{x}}} p_{\sigma,\tau}(\tilde{\boldsymbol{x}}|\boldsymbol{x},\tilde{\boldsymbol{y}},\boldsymbol{y}) d\boldsymbol{y} d\boldsymbol{x} \rangle d\tilde{\boldsymbol{y}} d\tilde{\boldsymbol{x}}$$

$$= \int_{\tilde{x}} \int_{\tilde{y}} \int_{x} \int_{y} p_{\tau}(\tilde{y})p_{0,\tau}(x|\tilde{y})p_{0,\tau}(y|x,\tilde{y})p_{\sigma,\tau}(\tilde{x}|x,\tilde{y},y)\langle \nabla_{\tilde{x}} \log p(\tilde{y}|\tilde{x};\theta), \nabla_{\tilde{x}} \log p_{\sigma,\tau}(\tilde{x}|x,\tilde{y},y)\rangle dydxd\tilde{y}d\tilde{x}$$

$$= \int_{\tilde{x}} \int_{\tilde{y}} \int_{x} \int_{y} p_{\sigma,\tau}(\tilde{x},x,\tilde{y},y)\langle \nabla_{\tilde{x}} \log p(\tilde{y}|\tilde{x};\theta), \nabla_{\tilde{x}} \log p_{\sigma,\tau}(\tilde{x}|x,\tilde{y},y)\rangle dydxd\tilde{y}d\tilde{x}$$

$$= \int_{\tilde{x}} \int_{\tilde{y}} \int_{x} \int_{y} p_{\sigma,\tau}(\tilde{x},x,\tilde{y},y)\langle \nabla_{\tilde{x}} \log p(\tilde{y}|\tilde{x};\theta), \nabla_{\tilde{x}} \log p_{\sigma,\tau}(\tilde{x},x,\tilde{y},y)\rangle dydxd\tilde{y}d\tilde{x}$$

$$= \int_{\tilde{x}} \int_{\tilde{y}} \int_{x} \int_{y} p_{\sigma,\tau}(\tilde{x},x,\tilde{y},y)\langle \nabla_{\tilde{x}} \log p(\tilde{y}|\tilde{x};\theta), \nabla_{\tilde{x}} \log(p_{\sigma}(\tilde{x}|x)p_{\tau}(\tilde{y}|y)p_{0,0}(x,y))\rangle dydxd\tilde{y}d\tilde{x}$$

$$= \int_{\tilde{x}} \int_{\tilde{y}} \int_{x} \int_{y} p_{\sigma,\tau}(\tilde{x},x,\tilde{y},y)\langle \nabla_{\tilde{x}} \log p(\tilde{y}|\tilde{x};\theta), \nabla_{\tilde{x}} \log p_{\sigma}(\tilde{x}|x)\rangle dydxd\tilde{y}d\tilde{x}$$

$$= \mathbb{E}_{p_{\sigma,\tau}(\tilde{x},x,\tilde{y},y)} \left[ \langle \nabla_{\tilde{x}} \log p(\tilde{y}|\tilde{x};\theta), \nabla_{\tilde{x}} \log p_{\sigma}(\tilde{x}|x)\rangle \right], \text{ and now we expand } L_{\text{DLSM}}(\theta):$$

$$L_{\text{DLSM}}(\theta) = \mathbb{E}_{p_{\sigma,\tau}(\tilde{x},x,\tilde{y},y)} \left[ \frac{1}{2}\|\nabla_{\tilde{x}} \log p(\tilde{y}|\tilde{x};\theta) + \nabla_{\tilde{x}} \log p_{\sigma}(\tilde{x}) - \nabla_{\tilde{x}} \log p_{\sigma}(\tilde{x}|x)\|^2 \right]$$

$$= \mathbb{E}_{p_{\sigma,\tau}(\tilde{x},\tilde{y})} \left[ \frac{1}{2}\|\nabla_{\tilde{x}} \log p(\tilde{y}|\tilde{x};\theta)\|^2 \right] - \mathbb{E}_{p_{\sigma}(\tilde{x},x,\tilde{y},y)} \left[ \langle \nabla_{\tilde{x}} \log p(\tilde{y}|\tilde{x};\theta), \nabla_{\tilde{x}} \log p_{\sigma}(\tilde{x}|x)\rangle \right] +$$

$$\mathbb{E}_{p_{\sigma,\tau}(\tilde{x},\tilde{y})} \left[ \langle \nabla_{\tilde{x}} \log p(\tilde{y}|\tilde{x};\theta), \nabla_{\tilde{x}} \log p_{\sigma}(\tilde{x})\rangle \right] + C_2, \text{ where}$$

$$C_2 = \mathbb{E}_{p_{\sigma}(\tilde{x},x)} \left[ \frac{1}{2}\|\nabla_{\tilde{x}} \log p_{\sigma}(\tilde{x})\|^2 \right] + \mathbb{E}_{p_{\sigma}(x,\tilde{x})} \left[ \frac{1}{2}\|\nabla_{\tilde{x}} \log p_{\sigma}(\tilde{x}|x)\|^2 \right] -$$

$$\mathbb{E}_{p_{\sigma}(\tilde{x},x)} \left[ \langle \nabla_{\tilde{x}} \log p_{\sigma}(\tilde{x}), \nabla_{\tilde{x}} \log p_{\sigma}(\tilde{x}|x)\rangle \right], \text{ is a constant, leading to the conclusion}$$

$$L_{\text{DLSM}}(\theta) = L_{\text{ELSM}}(\theta) - C_1 + C_2$$

Note that the domain of the integrals is $\{(\tilde{x},x,\tilde{y},y) : p_{\sigma,\tau}(\tilde{x},x,\tilde{y},y) \neq 0\}$ by the definition of expected value.

$\square$

**Remark.** Theorem 1 also holds for the clean label $y$, which can be shown as follows:

$$\mathbb{E}_{p_{\sigma}(\tilde{x},y)} \left[ \frac{1}{2}\|\nabla_{\tilde{x}} \log p(y|\tilde{x};\theta) - \nabla_{\tilde{x}} \log p_{\sigma}(y|\tilde{x})\|^2 \right]$$

$$= \mathbb{E}_{p_{\sigma}(\tilde{x},x,y)} \left[ \frac{1}{2}\|\nabla_{\tilde{x}} \log p(y|\tilde{x};\theta) + \nabla_{\tilde{x}} \log p_{\sigma}(\tilde{x}) - \nabla_{\tilde{x}} \log p_{\sigma}(\tilde{x}|x)\|^2 \right] + C.$$

## A.6 THE DETAILED EXPERIMENTAL CONFIGURATIONS

In this section, we elaborate on the implementation details and the experimental setups. In Section A.6.1, we describe the noise perturbation method and the sampling algorithm adopted in this work. In Sections A.6.2 and A.6.3, we present the training configurations for the experiments on the inter-twinning moon, Cifar-10, and Cifar-100 datasets. The code implementation for these experiments is provided in the Github repository (`https://github.com/chen-hao-chao/dlsm`).

### A.6.1 NOISE PERTURBATION AND SAMPLING ALGORITHM

**Sampling Algorithm.** In this work, we adopt the PC sampler (VE-SDE) identical to that in (Song et al., 2021b), which is a time-inhomogeneous sampling algorithm. The PC sampler (VE-SDE) is described in **Algorithm 1**. In **Algorithm 1**, a noise-conditioned score model $s(\tilde{x}; \phi, \sigma_t)$ is adopted, where $\tilde{x}$ is perturbed by noises sampled from Gaussian distributions using different standard deviations $\sigma_t$. The standard deviations of the noises, denoted as $\sigma_t = \sigma_{\min}(\frac{\sigma_{\max}}{\sigma_{\min}})^{\frac{t}{T}}$, are determined by the time step $t$ in **Algorithm 1**. We set $\sigma_{\min} = 0.01$ for all of our experiments. On the other hand, we set $\sigma_{\max} = 50$ for the Cifar-10 and Cifar-100 datasets, and $\sigma_{\max} = 10$ for the inter-twinning moom dataset. As for the total time steps, the value of $T$ is set to 1,000.

---

**Algorithm 1** The PC Sampler (VE-SDE)

---

1: $\tilde{\boldsymbol{x}} = \mathcal{N}(\mathbf{0}, \sigma_{\max}^2 \boldsymbol{I})$
2: **for** $t = T - 1$ **to** $0$ **do**
3:     /* Predictor */
4:     $\mathbf{z} \sim \mathcal{N}(\mathbf{0}, \boldsymbol{I})$
5:     $\tilde{\boldsymbol{x}}_t \leftarrow \tilde{\boldsymbol{x}}_t + (\sigma_{t+1}^2 - \sigma_t^2)\, s(\tilde{\boldsymbol{x}}_{t+1}; \phi, \sigma_{t+1}) + \sqrt{\sigma_{t+1}^2 - \sigma_t^2}\, \mathbf{z}$
6:     /* Corrector */
7:     $\mathbf{z} \sim \mathcal{N}(\mathbf{0}, \boldsymbol{I})$
8:     $\tilde{\boldsymbol{x}}_t \leftarrow \tilde{\boldsymbol{x}}_t + \epsilon_t\, s(\tilde{\boldsymbol{x}}_t; \phi, \sigma_t) - \sqrt{2\epsilon_t}\mathbf{z}$
9: **return** $\tilde{\boldsymbol{x}}_0$

---

**Training with Different Standard Deviations.** Since the PC sampler (VE-SDE) described in **Algorithm 1** requires the score model $s(\tilde{\boldsymbol{x}}; \phi, \sigma_t)$ conditioned on different values of standard deviation $\sigma_t$, the training objectives described in Eqs. (4) and (11) are modified accordingly. For the DSM loss described in Eq. (4), the modified loss $L_{\mathrm{DSM}}(\phi)$ is represented as:

$$\mathbb{E}_{\mathcal{U}_{[0,T]}(t)} \left[ \lambda_{\mathrm{DSM}}(t) \mathbb{E}_{p_{\sigma_t}(\tilde{\boldsymbol{x}}, \boldsymbol{x})} \left[ \| s(\tilde{\boldsymbol{x}}; \phi, \sigma_t) - \nabla_{\tilde{\boldsymbol{x}}} \log p_{\sigma_t}(\tilde{\boldsymbol{x}}|\boldsymbol{x}) \|^2 \right] \right], \tag{A3}$$

where $\mathcal{U}_{[0,T]}(t)$ is a uniform distribution on the interval $[0, T]$. In Eq. (A3), The additional coefficient $\lambda_{\mathrm{DSM}}(t)$ is multiplied with the original DSM loss for better training results (Song & Ermon, 2019). For the approximated DLSM loss described in Eq. (9), the modified loss $L_{\mathrm{DLSM}'}(\theta)$ is represented as:

$$\mathbb{E}_{\mathcal{U}_{[0,T]}(t)} \left[ \lambda_{\mathrm{DLSM}}(t) \mathbb{E}_{p_{\sigma_t, \tau}(\tilde{\boldsymbol{x}}, \boldsymbol{x}, \tilde{\boldsymbol{y}}, \boldsymbol{y})} \left[ \| \nabla_{\tilde{\boldsymbol{x}}} \log p(\tilde{\boldsymbol{y}}|\tilde{\boldsymbol{x}}; \theta, \sigma_t) + s(\tilde{\boldsymbol{x}}; \phi, \sigma_t) - \nabla_{\tilde{\boldsymbol{x}}} \log p_{\sigma_t}(\tilde{\boldsymbol{x}}|\boldsymbol{x}) \|^2 \right] \right], \tag{A4}$$

where $p(\tilde{\boldsymbol{y}}|\tilde{\boldsymbol{x}}; \theta, \sigma_t)$ is a noise-conditioned classifier. $\lambda_{\mathrm{DLSM}}(t)$ is the balancing coefficient for different $\sigma_t$. For the cross-entropy loss, the modified loss $L_{\mathrm{CE}}(\theta)$ is represented as:

$$\mathbb{E}_{\mathcal{U}_{[0,T]}(t)} \left[ \lambda_{\mathrm{CE}}(t) \mathbb{E}_{p_{\sigma_t, \tau}(\tilde{\boldsymbol{x}}, \tilde{\boldsymbol{y}})} [- \log p(\tilde{\boldsymbol{y}}|\tilde{\boldsymbol{x}}; \theta, \sigma_t)] \right], \tag{A5}$$

where $\lambda_{\mathrm{CE}}(t)$ is the balancing coefficient for different $\sigma_t$. In all of our experiments, $\lambda_{\mathrm{CE}}(t)$ is set to 1. The values of $\lambda_{\mathrm{DSM}}(t)$ and $\lambda_{\mathrm{DLSM}}(t)$ are customized according to different settings, and specified in Sections A.6.2 and A.6.3. Please note that $\lambda_{\mathrm{DSM}}(t)$, $\lambda_{\mathrm{DLSM}}(t)$, and $\lambda_{\mathrm{CE}}(t)$ are adopted to balance the loss functions for different $\sigma_t$, and are different from the balancing coefficient $\lambda$ in Eq. (11).

### A.6.2 THE EXPERIMENTS ON THE INTER-TWINNING MOON DATASET

**Dataset.** We perform our motivational experiments on the inter-twinning moon dataset provided by the `sklearn` library in `python`. In this dataset, the data points are sampled from two interleaving crescents. The data points of the upper crescent are generated based on $(\cos(x), \sin(x)), \forall x \in [0, \pi]$ and are labeled as $c_1$. On the other hand, the data points of the lower crescent are generated based on $(1 - \cos(x), 0.5 - \sin(x)), \forall x \in [0, \pi]$ and are labeled as $c_2$. Before training, the data points are first normalized such that their mean become 0 and then their coordinates are scaled by a factor of 20.

**Setups and Evaluation Method.** In this experiment, the scores $\nabla_{\tilde{\boldsymbol{x}}} \log p_\sigma(\tilde{\boldsymbol{x}})$ and $\nabla_{\tilde{\boldsymbol{x}}} \log p_\sigma(\tilde{\boldsymbol{x}}|\tilde{\boldsymbol{y}})$ are obtained by calculating Eq. (2). The likelihood scores are computed according to $\nabla_{\tilde{\boldsymbol{x}}} \log p_\sigma(\tilde{\boldsymbol{y}}|\tilde{\boldsymbol{x}}) = \nabla_{\tilde{\boldsymbol{x}}} \log p_\sigma(\tilde{\boldsymbol{x}}|\tilde{\boldsymbol{y}}) - \nabla_{\tilde{\boldsymbol{x}}} \log p_\sigma(\tilde{\boldsymbol{x}})$ in Eq. (5). As for the $D_P$ and $D_L$ metrics in Table 1, the errors between two score functions are measured using 1,225 data points uniformly sampled from the two-dimensional subspace $[-25, 25] \times [-40, 40]$. The scaling factor $\alpha$ adopted in method (b) is set to 10. The values of $\lambda_{\mathrm{DSM}}(t)$ and $\lambda_{\mathrm{DLSM}}(t)$ are both set to $\frac{1}{\sigma_t^4}$, and the value of $\lambda$ is set to 0.125.

**Training and Implementation Details.** The network architectures for the score model and the classifier are simple three-layer multilayer perceptrons (MLPs) with ReLu as the activation function. These networks are implemented using the `pytorch` library with `nn.Linear` layers that contain (128, 64, 32) neurons per layer for both of the score model and the classifier. The score model is trained using the Adam optimizer with a learning rate of $6.5 \times 10^{-4}$ and a batch size of 4,000 for 40,000 iterations. The classifier is trained using the Adam optimizer with a learning rate of $2.0 \times 10^{-5}$ and a batch size of 4,000 for 40,000 iterations.

Table A3: The experimental results on the inter-twining moon dataset, where the classifiers are trained with $L_{\text{CE}}$, $L_{\text{DLSM}'}$, and $L_{\text{Total}}$, respectively. The errors of the score functions for different methods are measured using $\mathbb{E}_{U(\tilde{\boldsymbol{x}})}[D_P(\tilde{\boldsymbol{x}}, \tilde{\boldsymbol{y}})]$ and $\mathbb{E}_{U(\tilde{\boldsymbol{x}})}[D_L(\tilde{\boldsymbol{x}}, \tilde{\boldsymbol{y}})]$, where $U(\tilde{\boldsymbol{x}})$ is a uniform distribution over a two-dimensional subspace, $\tilde{\boldsymbol{y}}$ represents the classes specified in the second row of the table, and $D_P$ and $D_L$ are defined in Eq. (7). The arrow symbol $\downarrow$ indicates that lower values correspond to better performances.

| | | $L_{\text{CE}}$ | | $L_{\text{DLSM}'}$ | | $L_{\text{Total}}$ | |
|---|---|---|---|---|---|---|---|
| $\tilde{\boldsymbol{y}}$ | | $c_1$ | $c_2$ | $c_1$ | $c_2$ | $c_1$ | $c_2$ |
| $\mathbb{E}_{U(\tilde{\boldsymbol{x}})}[D_P(\tilde{\boldsymbol{x}}, \tilde{\boldsymbol{y}})]$ | $\downarrow$ | 0.198 | 0.197 | **0.066** | 0.065 | **0.066** | **0.064** |
| $\mathbb{E}_{U(\tilde{\boldsymbol{x}})}[D_L(\tilde{\boldsymbol{x}}, \tilde{\boldsymbol{y}})]$ | $\downarrow$ | 0.190 | 0.181 | 0.053 | 0.053 | **0.052** | **0.052** |

### A.6.3 THE EXPERIMENTS ON THE CIFAR-10 AND CIFAR-100 DATASETS

**Dataset.**    We adopt the Cifar-10 and Cifar-100 datasets (Krizhevsky et al., 2009) for evaluating our design on image generation tasks. Cifar-10 consists of 60,000 $32 \times 32$ RGB images with 10 classes, where each class contains 6,000 images. Cifar-100 is similar to Cifar-10, except it has 100 classes, where each class contains 600 images. The images are first normalized to $[-1, 1]^d$ before training.

**Setups and Evaluation Method.**    The results on the CAS metric are evaluated using a classifier (ResNet-50 (He et al., 2016)) trained with the generated samples. This classifier is trained to minimize the cross-entropy loss with the Adam optimizer using a learning rate of $2.0 \times 10^{-4}$ with a batch size of 150. The results on the P / R / D / C metrics are calculated using the officially released code. The hyperparameter $k$ used in the P / R / D / C metrics is fixed to 3. The results on the FID and IS metrics are computed on 50,000 samples using the `tensorflow_gan` library in `python`. The value of $\lambda_{\text{DSM}}(t)$ is set to $\frac{1}{\sigma_t^2}$, while the value of $\lambda_{\text{DLSM}}(t)$ is set to 1. The value of $\lambda$ is set to 1.

**Training and Implementation Details.**    The network architecture for the score model is the same as (Song et al., 2021b). The network architecture for the classifier is a modified ResNet-18 (He et al., 2016) for Cifar-10 and a modified ResNet-34 for Cifar-100. The modification only involves an additional conditional branch for encoding the standard deviation $\sigma_t$. The score model is trained with the Adam optimizer using a learning rate of $2.0 \times 10^{-4}$ and a batch size of 128. The classifier is trained with the Adam optimizer using a learning rate of $2.0 \times 10^{-4}$ and a batch size of 100.

### A.7 ADDITIONAL EXPERIMENTS

In this section, we present a number of additional experimental results and provide discussions on them. We first report the results of our method using different values of $\lambda$ on Cifar-10 to demonstrate that $L_{\text{Total}}$ is less sensitive to the choices of $\lambda$. Next, we apply the scaling technique on both the base method and our method with different choices of $\alpha$ to demonstrate the influence of the hyperparameter $\alpha$ on the evaluation results of several metrics. Finally, we provide uncurated examples on the Cifar-10 and Cifar-100 datasets.

### A.7.1 ABLATION ANALYSIS ON THE INTER-TWINNING MOON DATASET

Table A3 compares the evaluation results on the $\mathbb{E}_{U(\tilde{\boldsymbol{x}})}[D_P(\tilde{\boldsymbol{x}}, \tilde{\boldsymbol{y}})]$ and $\mathbb{E}_{U(\tilde{\boldsymbol{x}})}[D_L(\tilde{\boldsymbol{x}}, \tilde{\boldsymbol{y}})]$ metrics when the classifiers are trained with $L_{\text{CE}}$, $L_{\text{DLSM}'}$, and $L_{\text{Total}}$, respectively. It is observed that the score errors $\mathbb{E}_{U(\tilde{\boldsymbol{x}})}[D_P(\tilde{\boldsymbol{x}}, \tilde{\boldsymbol{y}})]$ and $\mathbb{E}_{U(\tilde{\boldsymbol{x}})}[D_L(\tilde{\boldsymbol{x}}, \tilde{\boldsymbol{y}})]$ are the lowest when $L_{\text{Total}}$ is adopted. The experimental results therefore validate the effectiveness of the proposed loss $L_{\text{Total}}$.

### A.7.2 THE EXPERIMENTS ON THE BALANCING COEFFICIENT

Table A4 demonstrates the evaluation results on FID, IS, and P / R / D / C metrics using $L_{\text{Total}}$ with different values of $\lambda$. It is observed that the evaluation results only slightly fluctuate on the six metrics with different choices of $\lambda$. The results thus experimentally validate that $L_{\text{Total}}$ is less sensitive to the choice of $\lambda$.

Table A4: The experimental results of our method using $L_{\text{Total}}$ with different value of $\lambda$. The arrow symbols ↑ / ↓ represent that a higher / lower evaluation result correspond to a better performance.

| | | $\lambda = 2.0$ | $\lambda = 1.0$ | $\lambda = 0.5$ | $\lambda = 0.125$ |
|---|---|---|---|---|---|
| FID | ↓ | 2.33 | **2.25** | 2.36 | **2.25** |
| IS | ↑ | 9.90 | 9.90 | **9.98** | 9.79 |
| Precision | ↑ | 0.64 | **0.65** | 0.64 | 0.64 |
| Recall | ↑ | **0.62** | **0.62** | **0.62** | **0.62** |
| Density | ↑ | 0.94 | **0.96** | 0.95 | 0.94 |
| Coverage | ↑ | 0.79 | **0.81** | 0.79 | 0.78 |

Table A5: A comparison of the evaluation results of the base method, the scaling method with different $\alpha$, ours method, and ours + scaling method with different $\alpha$ on the Cifar-10 dataset. The arrow symbols ↑ / ↓ represent that a higher / lower evaluation result correspond to a better performance.

| | | Base | Scaling | | | | | | Ours | Ours + Scaling | | | | | |
|---|---|---|---|---|---|---|---|---|---|---|---|---|---|---|---|
| $\alpha$ | - | 1.0 | 0.5 | 3.0 | 5.0 | 10.0 | 20.0 | | 1.0 | 0.5 | 3.0 | 5.0 | 10.0 | 20.0 |
| FID | ↓ | 4.10 | 3.22 | 6.16 | 8.06 | 12.48 | 16.59 | | **2.25** | 2.57 | 3.96 | 6.40 | 11.06 | 14.92 |
| IS | ↑ | 9.08 | 9.50 | 9.14 | 9.38 | 9.37 | 9.23 | | 9.90 | 9.79 | **10.00** | 9.95 | 9.73 | 9.53 |
| Precision | ↑ | 0.67 | 0.65 | 0.70 | 0.72 | **0.75** | **0.75** | | 0.65 | 0.62 | 0.69 | 0.71 | 0.73 | 0.74 |
| Recall | ↑ | 0.61 | 0.61 | 0.56 | 0.53 | 0.49 | 0.44 | | 0.62 | **0.63** | 0.57 | 0.53 | 0.49 | 0.44 |
| Density | ↑ | 1.05 | 0.98 | 1.17 | 1.25 | **1.36** | **1.36** | | 0.96 | 0.87 | 1.15 | 1.23 | 1.28 | 1.30 |
| Coverage | ↑ | 0.80 | 0.78 | 0.78 | 0.77 | 0.75 | 0.67 | | **0.81** | 0.77 | 0.79 | 0.77 | 0.71 | 0.66 |
| (CW) Precision | ↑ | 0.51 | 0.43 | 0.63 | 0.67 | 0.70 | **0.73** | | 0.56 | 0.48 | 0.65 | 0.68 | 0.71 | 0.72 |
| (CW) Recall | ↑ | 0.59 | 0.62 | 0.51 | 0.46 | 0.42 | 0.37 | | 0.61 | **0.63** | 0.55 | 0.50 | 0.45 | 0.39 |
| (CW) Density | ↑ | 0.63 | 0.46 | 0.97 | 1.10 | 1.23 | **1.30** | | 0.76 | 0.58 | 1.05 | 1.16 | 1.23 | 1.26 |
| (CW) Coverage | ↑ | 0.60 | 0.47 | 0.69 | 0.69 | 0.66 | 0.61 | | 0.71 | 0.61 | **0.75** | 0.73 | 0.66 | 0.61 |

### A.7.3    THE EXPERIMENTS ON THE SCALING TECHNIQUE

Table A5 demonstrates the evaluation results of the base method, the scaling method using different values of $\alpha$, our method, and our method with the scaling technique using different values of $\alpha$ (i.e., 'ours + scaling') on the Cifar-10 dataset. It is observed that the performance of the 'scaling method' and 'ours + scaling' method on the fidelity metrics, i.e., precision and density, improves as the value of $\alpha$ grows. Conversely, the performance of the 'scaling method' and 'ours + scaling' method on the FID, recall, and coverage metrics degrades as the value of $\alpha$ grows. This results thus experimentally demonstrate the effects of the scaling technique on the performance of the base method and our method.

### A.7.4    UNCURATED EXAMPLES

Figs. A3, A4, A5, and A6 depict a few uncurated examples that qualitatively compare the effectiveness of different methods.

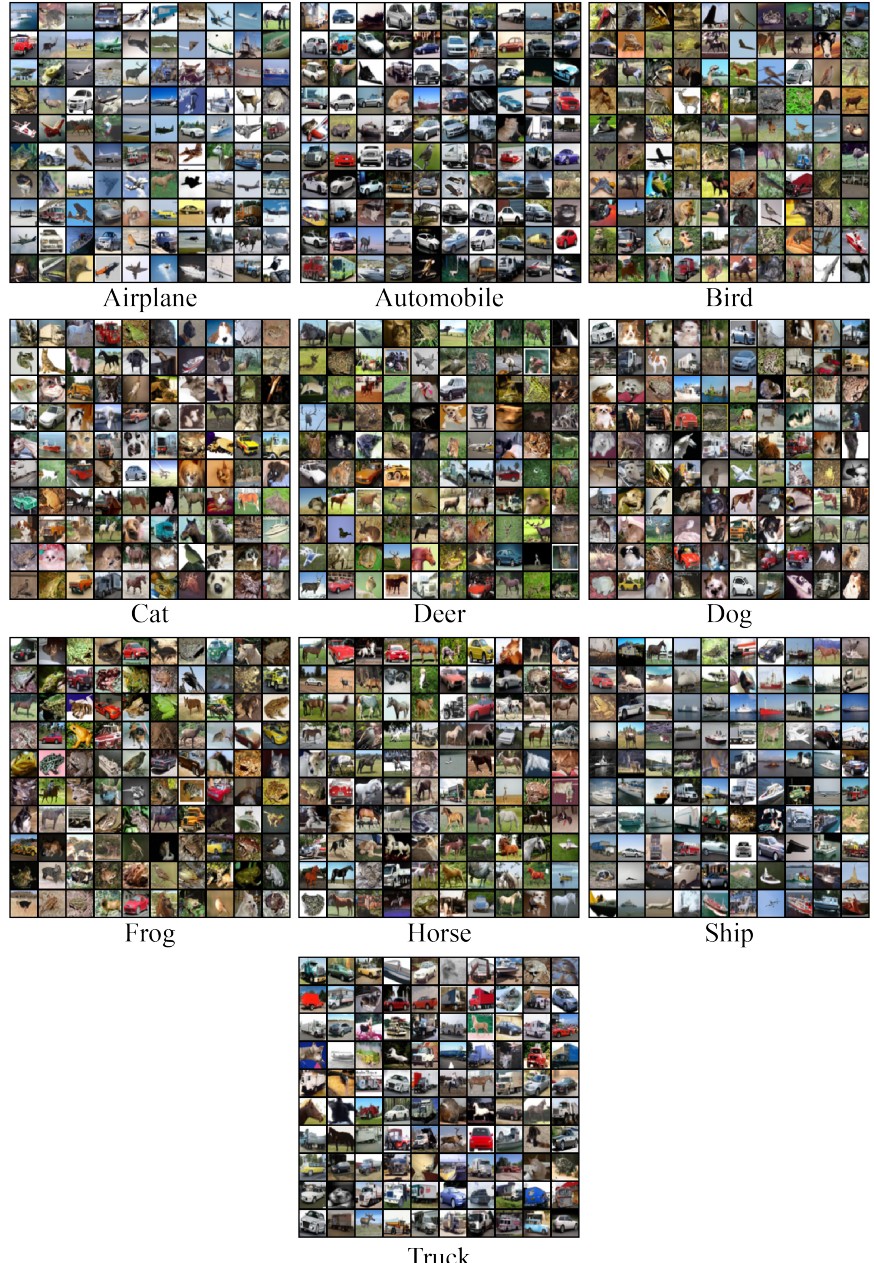

Figure A3: The uncurated examples generated using the base method on the Cifar-10 dataset.

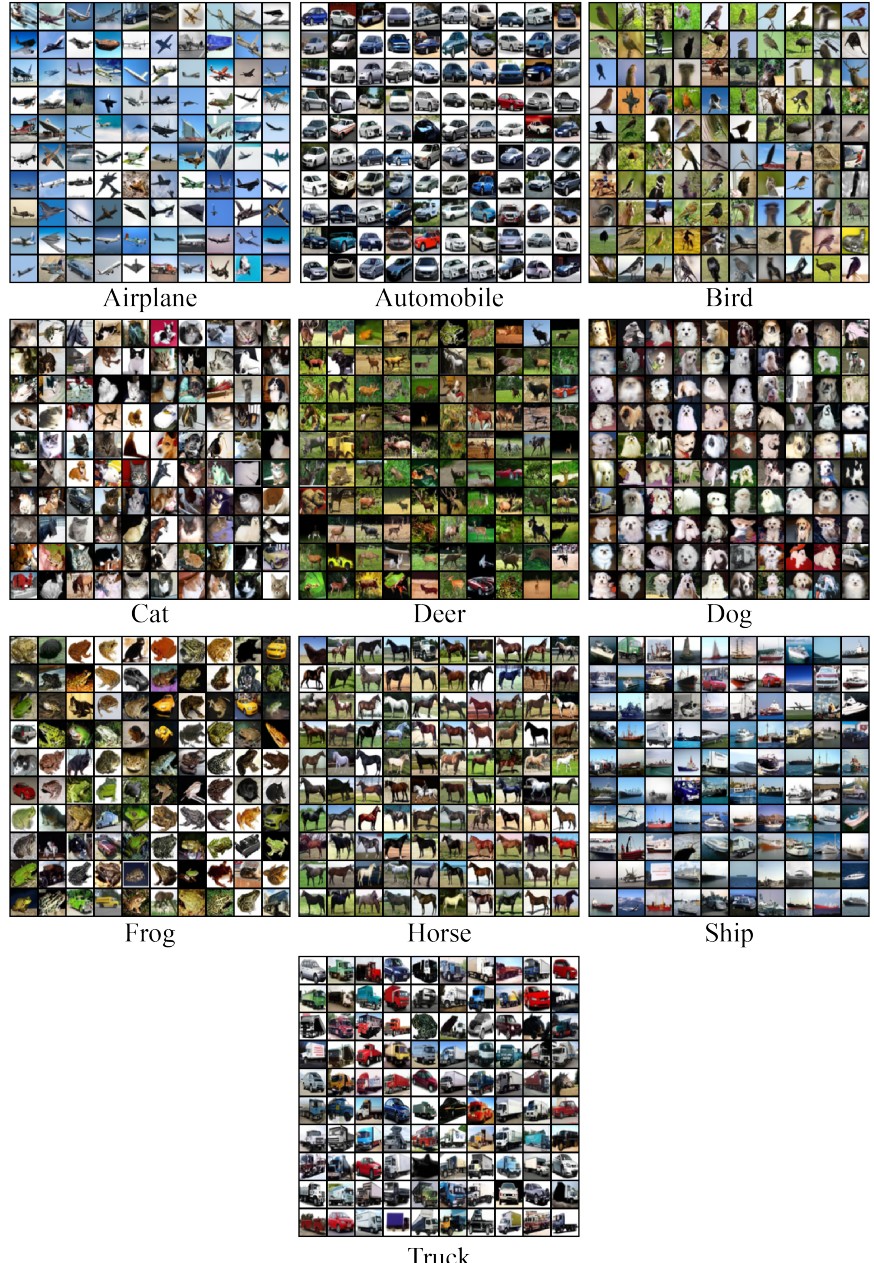

Figure A4: The uncurated examples generated using the scaling method on the Cifar-10 dataset.

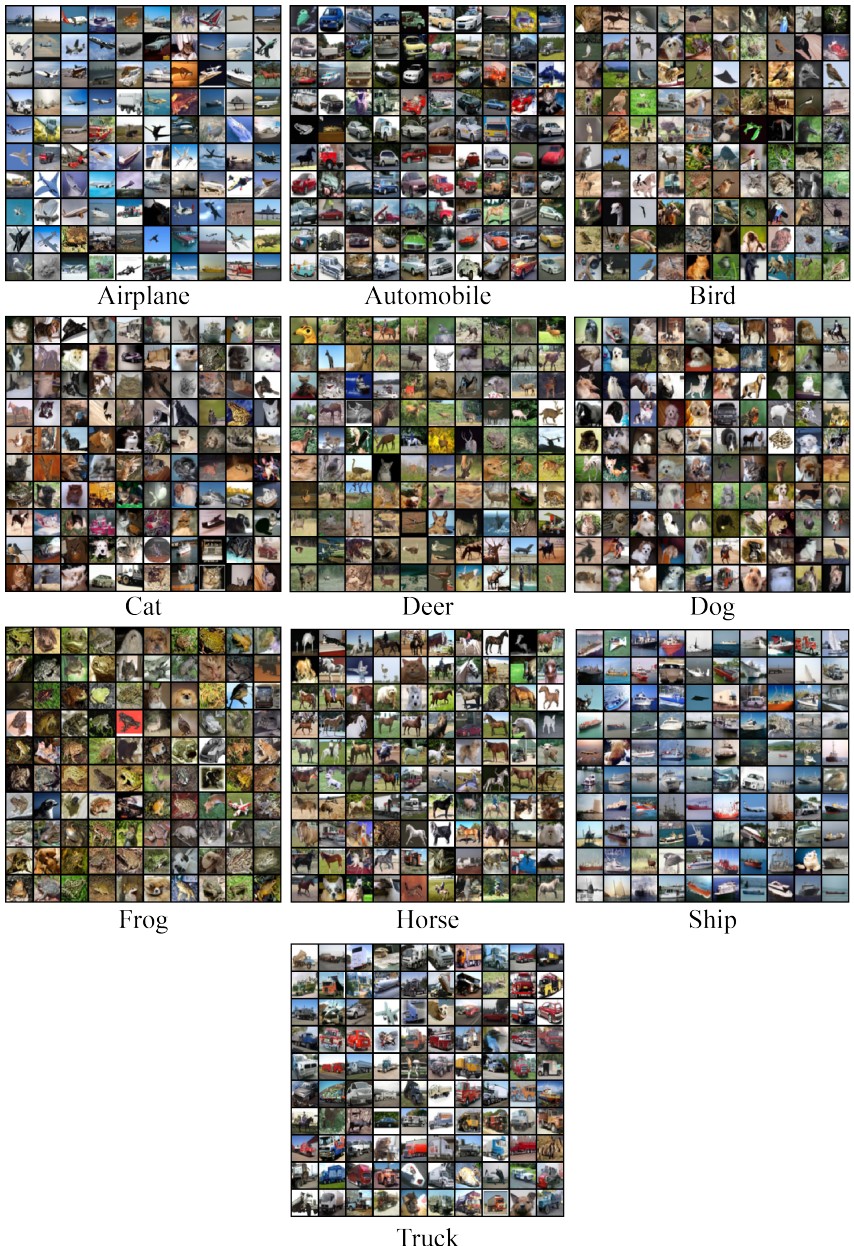

Figure A5: The uncurated examples generated using our method on the Cifar-10 dataset.

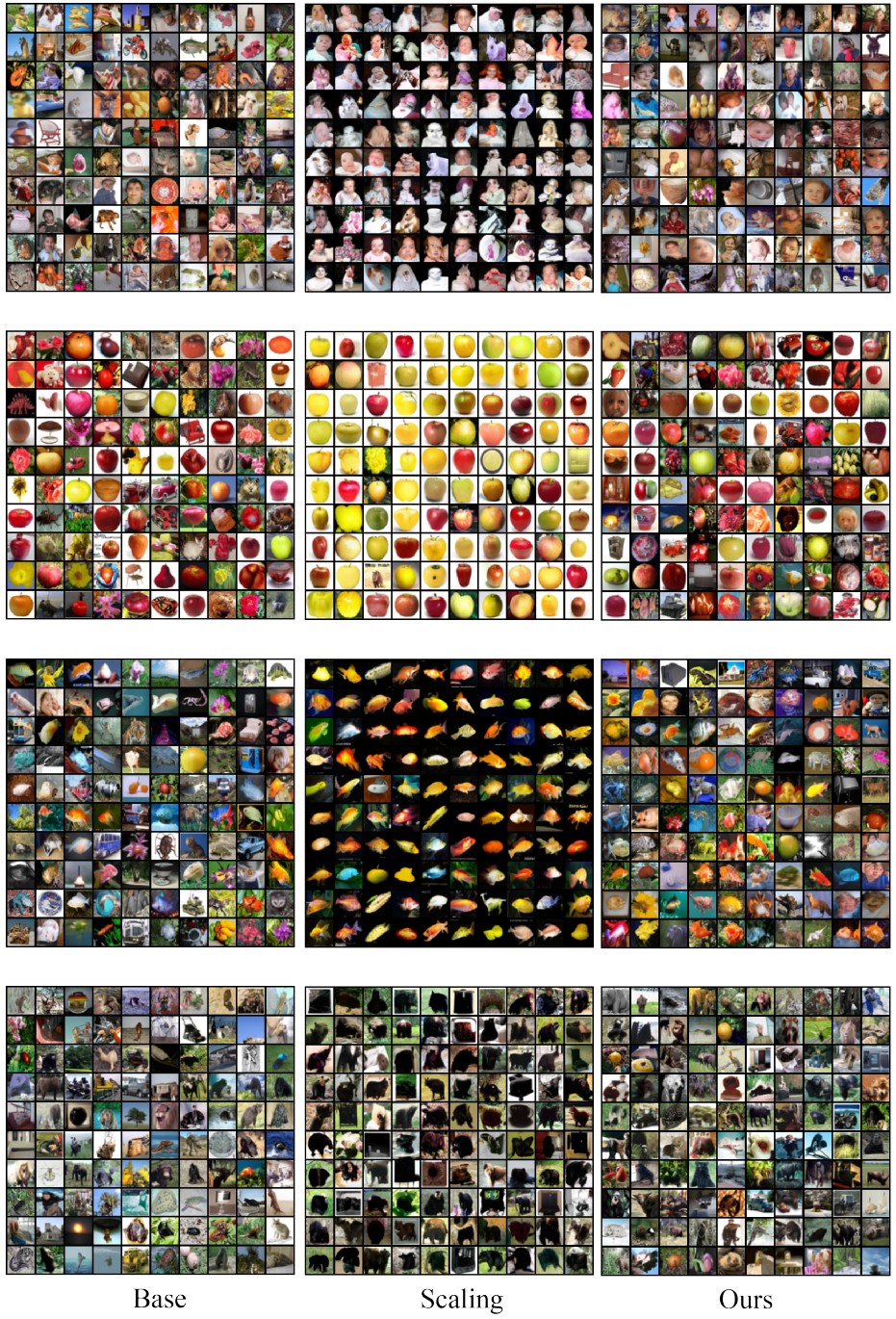

Base           Scaling           Ours

Figure A6: The uncurated examples of classes 'baby', 'apple', 'aquarium fish', and 'beargenerated' using the base method, the scaling method, and our method on the Cifar-100 dataset.

