# OpenReview forum: "Denoising Likelihood Score Matching for Conditional Score-based Data Generation"
_ICLR.cc/2022/Conference — ICLR 2022 Poster_

### Official Review · Reviewer_jmK6 · 2021-11-01

**Correctness:** 4
**Technical Novelty And Significance:** 4
**Empirical Novelty And Significance:** 3
**Recommendation:** 8
**Confidence:** 5

**Main Review:**

**Strengths**
- Sections 1 and 2 are very clear and exhibit strong understanding of the field and the most recent literature. Upon examination, the proof of (2) in A.2 appears to be correct.
- The motivating example, in Section 3 and Figure 1, is clearly presented, describes a relevant synthetic case, and the results demonstrate enough improvement to mean the paper's flow and narrative works well. The metrics $D_P$ and $D_L$ are also valuable in this more tractable setting.
- In Section 4, Equations (9) and (10) follow intuitively and are well explained. I have been through the proof of Theorem 1 in detail and I believe it is correct (there wasn't even a typo!). Figure 2 is also a useful/pertinent depiction of the training process.
- I believe all of the experiments presented in Section 5 are useful, clear, and well laid out.

**Weaknesses**
The principle weakness I can see in the paper is the lack of a breakdown of "inaccurate likelihood scores".  Although empirical evidence suggests the classifier is at fault, the paper only uses references for this motivation. There is no explanation as to how one might stratify these classifier score inaccuracies. As one example, does a stronger classifier provide better gradients for conditional generation? This line of research would potentially bolster the training procedure in Figure 2 by further guiding the involvement of the classifier.

- The use of "some extra information" at the start of Section 2.4 is casual and belies the importance of the class.
- In Section A.3, I believe "was $p_{\alpha}(\mathbf{\tilde{x}},\mathbf{\tilde{y}})$ not" should be "$p_{\alpha}(\mathbf{\tilde{x}},\mathbf{\tilde{y}})$ was not".
- At the end of page 5: "numerical results in Table 1 are" rather than  "numerical results in Table 1 is"
- The introduction of $p_{0,\tau}(\mathbf{x}|\mathbf{\tilde{y}})$ in the 5th line of the $G(\theta)$ derivation is slightly jarring given that the zero in the subscript is not used elsewhere or explicitly described.

Stylistic choices:
- Use of \left[ and \right] and latex would be nicer, but this is subjective.
- Same goes for the use of \lVert and \rVert for norms instead of \parallel (more for divergences).

**Summary Of The Paper:**

The authors present a new objective, denoising likelihood score-matching (DLSM), and training mechanism for conditional score-based data generation. The new method is motivated by poor conditional score estimates observed in low-dimensional examples for the standard, Bayes-theorem-based, conditional data generation procedure of score-based models. The new objective is equivalent to directly learning to parameterise the score of p(y|x) demonstrates is high-dimensional benchmark datasets also. FID and IS improve due to this method, with the largest improvements being class-wise, and there is an overall tendency to trade off generated data recall for precision.

**Summary Of The Review:**

Overall, I think this is a strong paper that would make a positive contribution to the conference and the current discussion surrounding SBMs.  The paper is very well presented and explained throughout, with appropriate figures and tables, as well as accurate derivations. Class conditional generation does appear to improve based on this method, and the complementary performances trade-offs (distribution precision) are broken down and discussed. Besides one line of research that could be included in a future paper, weaknesses are minor and/or aesthetic.

---

> ### Author Response · Authors · 2021-11-21
> **Response to Reviewer jmK6 (Part 1/1)**
>
> We appreciate the time and effort from the reviewer.
>
> ---
>
> **Q1.** The principle weakness I can see in the paper is the lack of a breakdown of "inaccurate likelihood scores". Although empirical evidence suggests the classifier is at fault, the paper only uses references for this motivation.
>
> **Q2.** There is no explanation as to how one might stratify these classifier score inaccuracies. As one example, does a stronger classifier provide better gradients for conditional generation? This line of research would potentially bolster the training procedure in Figure 2 by further guiding the involvement of the classifier.
>
> **A1 & A2.** We agree with the reviewer that a further investigation of the breakdown of inaccurate likelihood scores caused by the classifier would be an interesting future research direction. We also agree that an examination of the correlation between the classifier and the inaccurate likelihood scores could be a potential way to provide insights for guiding the training procedure.
>
> ---
> **Q3.**
> - The use of "some extra information" at the start of Section 2.4 is casual and belies the importance of the class. In Section A.3, I believe "was $p_{\alpha}(\tilde{\boldsymbol{x}},\tilde{\boldsymbol{y}})$ not" should be "$p_{\alpha}(\tilde{\boldsymbol{x}},\tilde{\boldsymbol{y}})$ was not".
> - At the end of page 5: "numerical results in Table 1 are" rather than "numerical results in Table 1 is "
> - The introduction of $p_{0, \tau}(\tilde{\boldsymbol{x}}|\tilde{\boldsymbol{y}})$ in the 5th line of the $G(\theta)$ derivation is slightly jarring given that the zero in the subscript is not used elsewhere or explicitly described.
>
> Stylistic choices:
> - Use of \left[ and \right] and latex would be nicer, but this is subjective.
> - Same goes for the use of \lVert and \rVert for norms instead of \parallel (more for divergences).
>
> **A3.** We appreciate the suggestions from the reviewer, and have fixed the typos, adjusted the derivations in Section A5, and modified the style of the equations. The new version of the paper has been updated, where the revised parts are highlighted in red.
>
> ---
>
> Finally, we would like to thank the reviewer for the thorough summary and the constructive advice.

---

### Official Review · Reviewer_UXBu · 2021-11-03

**Correctness:** 3
**Technical Novelty And Significance:** 2
**Empirical Novelty And Significance:** 1
**Recommendation:** 6
**Confidence:** 3

**Details Of Ethics Concerns:**

This paper does not provide any ethical concern

**Main Review:**

1.
I have a question on the derivation of Eq (A1). Particularly, authors suddenly introduce $Z(\tilde{x})$. It will be great if authors can explain how to main the equality, after the introduction of $Z$.

2.
It seems that the scores are mismatched across conditions. For example, the score on condition A cannot be matched to the score on condition B, which is quite natural. Given that it is good to see that the authors performed the posterior score-matching by separating the score-matching by conditions. At lease, Figure 1 shows not much difference between posterior SM and suggested models. Also, Table 1 suggests not much difference (or rather favorable to (c) posterior SM than the suggested models) between the posterior SM and the suggested model.

Then, what would be the gain by not just using posterior SM and by following the suggested models?

3.
I cannot find the clear definition of $L_{CE}$ which would be basically classifier, but it would be better to give a clear form of classifier with inputs and outputs.

4.
The experiment does not report negative loglikelihood, or bpd. Why don't you report NLL?

5.
There is no evaluation through the comparisons with the baseline models. As authors suggested that there are recent rush on the conditional modeling with diffusion approaches, why don't you find some and compare the performance with them?


**Summary Of The Paper:**

This paper presents a conditional diffusion(score-matching) model to tackle the score mismatch issue in the conditional generation scenario. This paper tests multiple alternatives in creating a conditional distribution through diffusion models, and this paper introduces a classifier assisted structure for the conditioning.

**Summary Of The Review:**

Niche paper to introduce a conditional generative model by the diffusion process

---

> ### Author Response · Authors · 2021-11-21
> **Response to Reviewer UXBu (Part 2/2)**
>
> **Q4.** The experiment does not report negative log likelihood, or bpd. Why don't you report NLL?
>
> **A4.** We consider that the FID, IS, CAS, Precision, Recall, Density and Coverage (P / R / D / C) metrics are sufficient to reflect the sampling quality. First of all, NLL is a metric for quantifying the ability of a generative model to approximate the target data distribution, and is very similar to FID, IS, P / R / D / C in terms of its functionality. Additionally, we use CAS to measure whether the generated samples bear representative class information.
>
> More comparisons between NLL and D / C metrics can be found in Section 4.1.2 in [1]. As mentioned in their paper:
> > … While NLL provides a single view of the generative model, the D&C metrics provide further diagnostic information. …
>
> The authors in [1] offered a discussion about the NLL metrics, and supported the above claim by a number of experimental evidence. As the D / C metrics can be adopted for a more comprehensive examination than NLL, we thus regard NLL as an optional choice for the evaluation metric.
>
> Based on these reasons, the experimental results with the adopted metrics are sufficiently capable of exhibiting the effectiveness of the proposed method.
>
> [1] Reliable Fidelity and Diversity Metrics for Generative Models, Muhammad Ferjad Naeem et al., ICML 2020
>
> ---
>
> **Q5.** There is no evaluation through the comparisons with the baseline models. As the authors suggested that there is a recent rush on the conditional modeling with diffusion approaches, why don't you find some and compare the performance with them?
>
> **A5.** The baseline methods of this paper are discussed in Section 3, which include (a) Base Method and (b) Scaling Method.  Method (a) was proposed in Section 2.1 of [2], and mentioned in Section 5 of [1] and Section 4.3 of [3]. On the other hand, the scaling technique adopted in (b) is proposed in Section 4.3 of [3], and described in Eqs. (4) and (5) of [2]. In addition, the comparison between the proposed method against methods (a) and (b) were provided in Tables 1 and 2.
>
> [1] Score-Based Generative Modeling through Stochastic Differential Equations, Yang Song et al., ICLR 2021 \
> [2] Plug & Play Generative Networks: Conditional Iterative Generation of Images in Latent Space, Anh Nguyen et al., CVPR 2017 \
> [3] Diffusion Models Beat GANs on Image Synthesis, Prafulla Dhariwal et al., ArXiv 2021
>
> ---
>
> We hope the above answers have adequately addressed the reviewer’s questions.

---

> > ### Comment · Reviewer_UXBu · 2021-11-25
> > **thanks**
> >
> > Thanks for the response and I updated my score to be on the accept side.

---

> ### Author Response · Authors · 2021-11-21
> **Response to Reviewer UXBu (Part 1/2)**
>
> We appreciate the time and effort from the reviewer.
>
> ---
>
> **Q1.** I have a question on the derivation of Eq. (A1). Particularly, authors suddenly introduce $Z$. It will be great if authors can explain how to maintain the equality, after the introduction of $Z$.
>
> **A1.** We appreciate your question, and have uploaded a revision providing more discussions in Section A.3.
>
> ---
>
> **Q2.**  It seems that the scores are mismatched across conditions. For example, the score on condition A cannot be matched to the score on condition B, which is quite natural. Given that it is good to see that the authors performed the posterior score-matching by separating the score-matching by conditions. At lease, Figure 1 shows not much difference between posterior SM and suggested models. Also, Table 1 suggests not much difference (or rather favorable to (c) posterior SM than the suggested models) between the posterior SM and the suggested model.
>
> **A2.** We would like to address the question of the reviewer as follows.
>
> First, we would like to clarify that the *score-mismatching issue* mentioned in our manuscript actually means the mismatch between the true posterior score and its estimated counterpart (i.e., For condition $\boldsymbol{y}\_1$, the true posterior score is $\nabla\_{\tilde{\boldsymbol{x}}} \log p(\tilde{\boldsymbol{x}}|\boldsymbol{y}\_1)$ and its estimation is $\nabla\_{\tilde{\boldsymbol{x}}} \log p(\tilde{\boldsymbol{x}}|\boldsymbol{y}\_1;\theta,\phi)$; for condition $\boldsymbol{y}\_2$, the true posterior score is $\nabla\_{\tilde{\boldsymbol{x}}} \log p(\tilde{\boldsymbol{x}}|\boldsymbol{y}\_2)$ and its estimation is $\nabla\_{\tilde{\boldsymbol{x}}} \log p(\tilde{\boldsymbol{x}}|\boldsymbol{y}\_2;\theta,\phi)$; etc.), as described in Section 1 of the manuscript:
>
> > … a classifier may suffer from a *score-mismatch issue*, which is the situation that the estimated posterior scores deviate from the true ones. ...
>
> Next, the posterior SM method, i.e, method (c), is only included in our work for analytical purposes to serve as a motivation for the derivation of the proposed DLSM loss, and is not directly comparable to methods (a), (b), and (d) due to the architectural difference. We would like to emphasize that method (c) actually utilizes a completely different paradigm as compared to methods (a), (b), as well as the proposed method (d). Therefore, method (c) is not supposed to be directly compared to our method, as pointed out in Section A4 of the appendix:
>
> > As described in Section 3, methods (a), (b), and (d) utilize a classifier $p(\tilde{\boldsymbol{y}}|\tilde{\boldsymbol{x}};\theta)$ and a score model $s(\tilde{\boldsymbol{y}};\phi)$ to estimate the posterior scores. In contrast, method (c) separately trains the posterior score models $s(\tilde{\boldsymbol{y}};\phi_1)$ and $s(\tilde{\boldsymbol{y}};\phi_2)$ for different class conditions. \
> ... \
> Therefore, in this work, we only compare the performance of methods (a), (b), and (d), and consider method (c) to be an orthogonal research direction.
>
> Finally, we would like to reiterate that the main theme of this paper is to highlight *score-mismatch issue* and introduce an effective conditional score model training method.
>
> Note that methods (a)~(d) represent: (a) Base Method, (b) Scaling Method, (c) Posterior SM Method, and (d) Our Method, respectively.
>
> ---
>
> **Q3.** I cannot find the clear definition of $L_{\mathrm{CE}}$ which would be basically classifier, but it would be better to give a clear form of classifier with inputs and outputs.
>
> **A3.** With regard to the question raised by the reviewer, we have edited the definition of $L_{\mathrm{CE}}$ in Section 2.4 of the manuscript. The edited portions are highlighted in red.  Furthermore, we would like to point out that $L_{\mathrm{CE}}$ is defined as the expected cross-entropy loss $\mathbb{E}\_{p\_{\sigma, \tau}(\tilde{\boldsymbol{x}},\tilde{\boldsymbol{y}})}[-\log p(\tilde{\boldsymbol{y}}|\tilde{\boldsymbol{x}};\theta)]=\mathbb{E}\_{p\_{\sigma}(\tilde{\boldsymbol{x}})}[\mathrm{CE}(p\_{\sigma, \tau}(\tilde{\boldsymbol{y}}|\tilde{\boldsymbol{x}}), p(\tilde{\boldsymbol{y}}|\tilde{\boldsymbol{x}};\theta))]$, where $\mathrm{CE}(p(\tilde{\boldsymbol{y}}|\tilde{\boldsymbol{x}}), p(\tilde{\boldsymbol{y}}|\tilde{\boldsymbol{x}};\theta))$ denotes the cross-entropy between the likelihood density $p\_{\sigma, \tau}(\tilde{\boldsymbol{y}}|\tilde{\boldsymbol{x}})$ and the classifier $p(\tilde{\boldsymbol{y}}|\tilde{\boldsymbol{x}};\theta)$.
>
> ---

---

### Official Review · Reviewer_2i3W · 2021-11-05

**Correctness:** 3
**Technical Novelty And Significance:** 4
**Empirical Novelty And Significance:** 4
**Recommendation:** 8
**Confidence:** 4

**Main Review:**

**Strengths of the paper**
In general, the paper's contributions are clear, and I also consider that the results are essential for several reasons:


First, the paper highlights a previously less-recognized problem in the context of conditional generations of diffusion-based generative models. Moreover, the authors provide sufficient analysis to help readers understand the problem and potential drawbacks of commonly accepted methods. In particular, the analysis to visualize the difference between the MLE-based training and the proposed method is very interesting to read.


Second, the paper introduces a novel training objective to tackle the problem above, called Denoising Likelihood Score Matching (DLSM). Recently, the interest in diffusion-based generative models has increased rapidly. Consequently, the interest in controlling the generation process of such models has also increased. Regarding this, I consider that the proposed method has huge applicability and will contribute to the ML communities.


Third, the authors have devoted themselves to discussing practical techniques for applications of the proposed method. For example, the authors discuss that the effect of combining two losses, i.e., MLE and approximate DLSM, for training classifiers. In particular, the paper discusses the potential roles of two losses on the performance of the conditional generations throughout ablation studies. Furthermore, the authors provide additional analysis about the scaling method, including how they contribute to high-precision samples.

**Weaknesses of the paper**
However, the following aspects of the paper can be improved.


First, clarifying three objectives seems necessary, i.e., ELSM, DLSM, and approximate DLSM. In my opinion, when the pre-trained score is plugged into the DLSM objective, it is no longer DLSM loss. However, the current version hasn't sufficiently emphasized the difference, so I found that the current version can be potentially misleading. Moreover, the provided proofs and relevant theoretical discussions are about ELSM and DLSM, not approximate. As a result, readers infer the behaviors of the approximate DLSM only by empirical results. Thus, it is important to distinguish between DLSM and approximate DLSM.


Second, while the experiments are well-thought and easy to follow in general, some experimental results are unclear. Moreover, additional experiments seem required. For instance, it is unclear how FID and IS are evaluated. At first glance, I understood that the FID/IS are evaluated per class (and their averaged value are reported). However, the paper describes that they are evaluated unconditionally. If the same model was used, the unconditional generators should have been the same for the baselines and the proposed method. If it wasn't, the description of the experiments needed to be updated. Another concern is that it is unclear when the combined loss is used and when it isn't during the experiments. In particular, it is unclear if "ours" in Table 1 corresponds to the approximate DLSM or the combined loss since the rest of the experiments have used the combined loss. I consider it important to clarify Table 1 and potentially include the results of models trained with the combined loss. Similarly, it would be interesting to see the results of CIFAR-10/100 experiments without MLE-losses.


Third, the presentation of some backgrounds can be improved. In particular, it seems like that the current submission introduces the discrete-time Langevin dynamics, which is time-homogeneous, as a building block of the diffusion-based generative models. However, the paper follows the recent works, specifically time-inhomogeneous diffusions, which have made significant distinctions of the recent works against the time-homogeneous ones in several perspectives; for example, practicality in the context of generative models.

**Minor comments**
- I found that matching the y-axis scale in Figs 4 and A.1. will help interpret the results.
- Optionally, I found it beneficial to run 1-dimensional experiments as done in Sec 3 but present results similar to Fig A.1. In my understanding, the classifiers trained with the approximate DLSM will underestimate the gradient (thus smoother) in comparison to the DLSM (I may be wrong). On the other hand, the MLE-based training results in unnecessarily sharper. Plotting the learned gradients and the ground truth will be helpful to explain why combining two losses makes sense.
- In my understanding, $\tilde{y}$ may not be necessary to derive DLSM. Could you explain why noisy $\tilde{y}$ is necessary?

**Summary Of The Paper:**

The paper points up a previously underappreciated problem in training classifiers in the context of conditional generations of diffusion-based generative models. The authors propose a novel objective for training classifiers to tackle the problems. Informally, the generation process of the diffusion-based generative models can be described by repeatedly applying an update rule with initial values: $\tilde{x} ← \tilde{x} + \nabla_{\tilde{x}} \log p_{\textrm{model}}(\tilde{x}) + \sigma \epsilon$ where $\epsilon \sim N(0,I)$ and the initial points are sampled from a prior distribution such as standard Normal distributions. Similarly to unconditional generations, one can perform conditional generations, e.g., $p(\tilde{x} | y)$ by using the following update rule: $\tilde{x} ← \tilde{x} + \nabla_{\tilde{x}} \log p_{\textrm{model}}(\tilde{x}) + \nabla_{\tilde{x}} \log p_{\textrm{model}}(y | \tilde{x}) + \sigma \epsilon$. Note that $\nabla_{\tilde{x}} \log p(\tilde{x} | y) = \nabla_{\tilde{x}} \log p_{\textrm{model}}(\tilde{x}) + \nabla_{\tilde{x}} \log p_{\textrm{model}}(y | \tilde{x})$. Thus, for a given pre-trained generative model, one needs to learn $\log p_{\textrm{model}}(y | \tilde{x})$ (or its gradient wrt $\tilde{x}$. Maximum likelihood training (MLE), i.e., minimizing cross-entropy loss, is commonly used. Inevitably, the qualities of the gradient $\nabla_{\tilde{x}} \log p_{\textrm{model}}(y | \tilde{x})$ will determine the qualities of the conditional generations.


First, the paper emphasizes that MLE-training of classifiers results in non-smooth gradient landscape wrt input; thus, the resulting gradients negatively affect the generation qualities. More precisely, with MLE, the learned $\log p_{\textrm{model}} (y|\tilde{x})$ is high, while its gradient wrt $\tilde{x}$ isn't necessarily close to the ground truth. In the paper, the authors refer to this phenomenon as *a score mismatch issue.* The authors analyze the mismatch issue with toy experiments and demonstrate its negative effects on generation qualities compared to the ground truth.


Second, to resolve the *score mismatch issue*, the paper proposes a new objective function, called Explicit Likelihood Score-Matching loss (ELSM), in which the mean squared errors between $\nabla_{\tilde{x}} \log p_{\textrm{model}} (y|\tilde{x})$ and $\nabla_{\tilde{x}} \log p_{\textrm{data}} (y|\tilde{x})$ is minimized. Here, due to the inaccessibility $\nabla_{\tilde{x}} \log p_{\textrm{data}} (y|\tilde{x})$, the authors reduces the ELSM loss to another objective function, named *Denoising Likelihood Score-Matching* (DLSM), similarly to denoising score matching derivation. In DLSM, $\nabla_{\tilde{x}} \log p_{\textrm{model}} (y|\tilde{x})$ is trained to match to $\nabla_{\tilde{x}} \log p_{\textrm{data}} (\tilde{x}) - \nabla_{\tilde{x}} \log p_{\textrm{data}} (\tilde{x} | x)$. Acknowledging that $\nabla_{\tilde{x}} \log p_{\textrm{data}} (\tilde{x})$ is still unattainable, the authors propose to minimize approximate DLSM loss, where $\nabla_{\tilde{x}} \log p_{\textrm{data}} (\tilde{x})$ is substituted by the pre-trained scores $\nabla_{\tilde{x}} \log p_{\textrm{model}} (\tilde{x})$.


Then, the authors demonstrate that by toy experiments, training classifiers with the approximate DLSM improves the conditional generation qualities compared to MLE-based training. Moreover, the authors mention that approximate DLSM-based training can be unstable in high-dimensional datasets and show that training classifiers by minimizing combined loss of approximate DLSM and cross-entropy can further improve the generation qualities.


Lastly, the paper demonstrates the effectiveness of the proposed methods by evaluating conditional generation qualities on CIFAR-10 and CIFAR-100 datasets.

**Summary Of The Review:**

In general, the paper's contributions are clear; the proposed methods are well-motivated and well-discussed. In addition, I also found that the paper has a well-organized structure so that it is clear to understand the proposed method and other practical techniques to improve training. Thus, I'm inclined to accept the paper. However, I found that several aspects of the submission can be improved. I hope that the aforementioned weak points are well addressed.

---

> ### Author Response · Authors · 2021-11-21
> **Response to Reviewer 2i3W (Part 2/2)**
>
> **Q4.** Third, the presentation of some backgrounds can be improved. In particular, it seems like that the current submission introduces the discrete-time Langevin dynamics, which is time-homogeneous, as a building block of the diffusion-based generative models. However, the paper follows the recent works, specifically time-inhomogeneous diffusions, which have made significant distinctions of the recent works against the time-homogeneous ones in several perspectives; for example, practicality in the context of generative models.
>
> **A4.** We are thankful for the constructive advice from the reviewer, and have revised parts of the descriptions about the sampling methods in Section 2.1 and Section A6.1. In Section 2.1 of the updated version, we provide a clarification of the difference between the time-homogeneous and time-inhomogeneous diffusions. In Section A6.1, we provide detailed descriptions about the VE-SDE sampler, and explain how to reformulate the loss functions when the VE-SDE sampler is adopted. We hope that the descriptions in the updated version addresses the concern of the reviewer.
>
> ---
>
> **Q5.** In my understanding, $\tilde{\boldsymbol{y}}$ may not be necessary to derive DLSM. Could you explain why noisy $\tilde{\boldsymbol{y}}$ is necessary?
>
> **A5.** We appreciate the interesting question about the perturbed labels $\tilde{\boldsymbol{y}}$ raised by the reviewer. The reason for introducing $\tilde{\boldsymbol{y}}$ in the DLSM loss is to ensure the data points with condition $\boldsymbol{y}\notin\\{\boldsymbol{y}^{(i)}\\}\_{i=1}^m$ can be sampled.
>
> In other words, if DLSM is not derived using the perturbed label $\tilde{\boldsymbol{y}}$, only the data points conditioned on $\boldsymbol{y}\in\\{\boldsymbol{y}^{(i)}\\}\_{i=1}^m$ can be generated since $p\_{\sigma,0}(\boldsymbol{y}\vert\tilde{\boldsymbol{x}})=0$ for all $\boldsymbol{y}\notin\\{\boldsymbol{y}^{(i)}\\}\_{i=1}^m$.
>
> Therefore, the unperturbed version can be viewed as a special case of the derived DLSM.
>
> ---
>
> Finally, we would like to thank the reviewer for the thorough summary and the constructive advice.

---

> ### Author Response · Authors · 2021-11-21
> **Response to Reviewer 2i3W (Part 1/2)**
>
> We would like to first thank the reviewer for thoroughly summarizing our paper. We think the summary could be very helpful for the future readers.
>
> ---
>
> **Q1.** First, clarifying three objectives seems necessary, i.e., ELSM, DLSM, and approximate DLSM. In my opinion, when the pre-trained score is plugged into the DLSM objective, it is no longer DLSM loss. However, the current version hasn't sufficiently emphasized the difference, so I found that the current version can be potentially misleading. Moreover, the provided proofs and relevant theoretical discussions are about ELSM and DLSM, not approximate. As a result, readers infer the behaviors of the approximate DLSM only by empirical results. Thus, it is important to distinguish between DLSM and approximate DLSM.
>
> **A1.** We appreciate the suggestions from the reviewer, and have revised the descriptions about DLSM in Section 4.1. The revised parts are highlighted in red. In the updated version, we define $L_\mathrm{DLSM'}(\theta)=\mathbb{E}\left[p_{\sigma,\tau}(\tilde{\boldsymbol{x}}, \boldsymbol{x}, \tilde{\boldsymbol{y}}, \boldsymbol{y})[ \frac{1}{2} \lVert \nabla_{\tilde{\boldsymbol{x}}} \log p(\tilde{\boldsymbol{y}}|\tilde{\boldsymbol{x}};\theta) + s(\tilde{\boldsymbol{x}};\phi) - \nabla_{\tilde{\boldsymbol{x}}} \log p_{\sigma}(\tilde{\boldsymbol{x}}| \boldsymbol{x} )\rVert^2 ]\right]$ as the approximated DLSM loss to distinguish itself from $L_{\mathrm{DLSM}}$.
>
> ---
>
> **Q2.** Second, while the experiments are well-thought and easy to follow in general, some experimental results are unclear. Moreover, additional experiments seem required. For instance, it is unclear how FID and IS are evaluated. At first glance, I understood that the FID/IS are evaluated per class (and their averaged value are reported). However, the paper describes that they are evaluated unconditionally. If the same model was used, the unconditional generators should have been the same for the baselines and the proposed method. If it wasn't, the description of the experiments needed to be updated.
>
> **A2.** We appreciate the suggestions from the reviewer, and have revised the descriptions about the evaluation methods in Section 5.2. The revised parts are highlighted in red as well. The reported FID / IS metrics in Tables 2, A3, A4 are computed on the conditional samples. The evaluation method is explained as follows:
>
> > Suppose a dataset contains $m_i$ images for each class $i$.
> > - (1) First, $m_i$ samples are conditionally generated for each class $i$.
> > - (2) Then, FID / IS are computed using all $\sum_{i=1}^c m_i$ samples from the dataset and the $\sum_{i=1}^c m_i$ conditionally generated samples, where $c$ denotes the total number of classes.
>
> Please note that the same evaluation procedure is also used to compute the P / R / D / C and CAS metrics.
>
> ---
>
> **Q3.** Another concern is that it is unclear when the combined loss is used and when it isn't during the experiments. In particular, it is unclear if "ours" in Table 1 corresponds to the approximate DLSM or the combined loss since the rest of the experiments have used the combined loss. I consider it important to clarify Table 1 and potentially include the results of models trained with the combined loss. Similarly, it would be interesting to see the results of CIFAR-10/100 experiments without MLE-losses.
>
> **A3.** We appreciate the question from the reviewer.  In Table 1, “(d) ours” method is optimized according to the combined loss $L_{\mathrm{Total}}$.  We have enhanced the descriptions in Section 3 and highlighted the revised part in red.
>
> In addition, since the readers may be interested in the ablation analysis, we added a short discussion in Section A7.1 for comparing the score errors ($D_P$, $D_L$) when the classifier is optimized according to three different losses (i.e., $L_{\mathrm{CE}}$, $L_{\mathrm{DLSM’}}$, and $L_{\mathrm{Total}}$) on the inter-twining moon dataset. The results demonstrated that the expectations of $D_{P}$ and $D_{L}$ are lower when $L_{\mathrm{Total}}$ is adopted. Please refer to Table A3 for the detailed information.

---

### Official Review · Reviewer_LMQh · 2021-11-06

**Correctness:** 3
**Technical Novelty And Significance:** 3
**Empirical Novelty And Significance:** 3
**Recommendation:** 8
**Confidence:** 4

**Details Of Ethics Concerns:**

No concerns.

**Main Review:**

Some disadvantages

1) The loss is quite noisy (see Figure 5), and the authors do not explain why.
I personally think that the reason is because they approximate p(y|x) with a mixture of Gaussians with a small variance and there is the same effect with smoothing as in case of the standard score matching.
Probably we should start training using larger variance values and gradually decrease the variance during the training.

2) I think that the motivation of the paper has too many somewhat irrelevant details. The authors compare different models, trained under different conditions, and conlude that the problem is in the loss. It seems that it was enough to say that it is difficult to approximate the gradients, so it is necessary to explicitly add them to the loss.

3) Models (a) and (c) are different (in the first case it is a single model, in the second case the authors use different models for different classes). The first model minimizes cross-entropy loss, the second model optimizes the score matching function.
It is not entirely clear why the authors believe that the model (a) works worse than the model (c)  because of the loss? The models are also different. Maybe this is the main reason?

4) In the same section with the motivation I propose to add the cross-entropy loss between p(y|x) and p(x|y), and not just the loss between their gradients (maybe the model (a) also approximates them poorly). The question immediately arises, why a good cross-entropy loss does not guarantee a good approximation of the gradients?

5) Why do the authors train the classifier p(y | x, \theta) and the score model (x, \phi) independently, not end-to-end?

6) Why is their loss noisy in practice? How much does the value of \tau (noise variance) affect the stability of the loss and the results? Why don't the authors do the same trick with \tau as with \sigma (that is, they don't train several different models with different values of \tau)?

7) Experiments are done only on cifar-10/cifar-100, which is not very convincing, of course.

**Summary Of The Paper:**

The authors suggest how to apply the idea of score matching to the problem of conditional generation. They introduce a new loss, which is essentially a score matching loss for the conditional distribution p(y| x). They demonstrate how to effectively calculate it without calculating p(y|x) explicitly, which seems to be the main contribution of the paper. Then the authors demonstrate how to train the model for the conditional generation in practice: it is necessary to add to the proposed loss the usual cross-entropy loss, because the loss that they came up with is noisy (this can be seen in Figure 5). They got better results compared to baselines for cifar-10 and cifar-100.

**Summary Of The Review:**

In general, the paper is well written, it has a clear and logical structure, it is easy to read.
In the introduction, only clearly highlighted contributions are missing.

The proposed criteria can be useful when training models for conditional generation.
The authors did rather detailed discussion of different variants of the score matching procedure.

============

I thank the authors for their responses. The updated version of the paper, with additional information here and there, improves the readability and the overall narrative. Given the authors responses now I am convinced in the concept proposed by the authors, so I can raise my score.

---

> ### Author Response · Authors · 2021-11-21
> **Response to Reviewer LMQh (Part 3/3)**
>
>
> **Q5.** Why do the authors train the classifier $p(\tilde{\boldsymbol{y}}|\tilde{\boldsymbol{x}};\theta)$ and the score model $s(\tilde{\boldsymbol{x}};\phi)$ independently, not end-to-end?
>
> **A5.** End-to-end training is not beneficial since the optimization of $\theta$ is dependent on $\phi$, while the optimization of $\phi$ does not depend on $\theta$. To be more specific, if the models are trained in an end-to-end manner, a classifier could fail to learn the likelihood scores when minimizing the DLSM loss based on a constantly changing posterior score model $s(\tilde{\boldsymbol{x}};\phi)$, while the posterior score model does not benefit from the joint training with the classifier $p(\tilde{\boldsymbol{y}}|\tilde{\boldsymbol{x}};\theta)$. Therefore, the proposed two-stage training procedure is a more reasonable and efficient approach when it is compared to the end-to-end training method.
>
> ---
>
> **Q7.** Experiments are done only on cifar-10/cifar-100, which is not very convincing, of course.
>
> **A7.** The main purpose of this work is to highlight the existence of the score-mismatch issue and provide the theoretical solution to it, instead of demonstrating the application performance.  In our experiments, we were able to demonstrate that this issue exists even in a low-dimensional example (E1), and image-based datasets such as Cifar-10 (as those adopted in [1-3]) and Cifar-100 (E2,E3). Therefore, the existing experimental results are adequately sufficient to justify the arguments made in this paper.
> - (E1) inter-twining moon dataset in Section 3 and Section 5.3.
> - (E2) cifar-10 dataset in Section 5.2.
> - (E3) cifar-100 dataset in Section 5.2.
>
> [1] Generative Modeling by Estimating Gradients of the Data Distribution, Yang Song et al., NeurIPS 2019 \
> [2] Denoising Diffusion Probabilistic Models, Jonathan Ho et al., NeurIPS 2020 \
> [3] Score-Based Generative Modeling through Stochastic Differential Equations, Yang Song et al., ICLR 2021
>
> ---
>
> We hope the above answers have adequately addressed the reviewer’s questions.

---

> ### Author Response · Authors · 2021-11-21
> **Response to Reviewer LMQh (Part 2/3)**
>
> **Q3.** Models (a) and (c) are different (in the first case it is a single model, in the second case the authors use different models for different classes). The first model minimizes cross-entropy loss, the second model optimizes the score matching function. It is not entirely clear why the authors believe that the model (a) works worse than the model (c) because of the loss? The models are also different. Maybe this is the main reason?
>
> **A3.** We did not claim in the paper that method (a) works worse than method (c) is only because of the loss. Instead, we hypothesized that the loss is the key factor.
>
> We would like to first clarify that method (c) is only included for analytical purposes to serve as a motivation for the derivation of the DLSM loss, and is not directly comparable to methods (a), (b), and (d) due to the architectural difference. When we introduced method (c) in Section 3, we stated that:
> > ... since this method (i.e., method (c)) requires network architectures different from those used in methods (a), (b), and (d), it is only adopted for analytical purposes. ...
>
> Furthermore, in Section A4 of the supplementary material, we also stated that:
> > ... Therefore, in this work, we only compare the performance of methods (a), (b), and (d), and consider method (c) to be another orthogonal research direction.
>
> Since the performance difference between methods (a) and (c) may result from the loss or the network architecture, the final performance can be influenced by either one or a compound of them. We hypothesized that the loss is the key factor and hence, in Section 3, we stated that:
> >... the adoption of the score-matching objective may potentially be the key factor to the success of method (c). ...
>
> The experimental results in Table 1 provide empirical support for this hypothesis, and the results in Table 2 verify that the addition of the score-matching loss indeed contributes to the improvement of the performance on real-world datasets.
>
> ---
>
> **Q4.** In the same section with the motivation I propose to add the cross-entropy loss between $p(\boldsymbol{y}|\boldsymbol{x})$ and $p(\boldsymbol{x}|\boldsymbol{y})$, and not just the loss between their gradients (maybe the model (a) also approximates them poorly). The question immediately arises, why a good cross-entropy loss does not guarantee a good approximation of the gradients?
>
> **A4.** First, including the cross-entropy loss on $p(\boldsymbol{x}|\boldsymbol{y})$ is infeasible in practice. To be more specific, the main reason of using denoising score-matching to approximate the scores, and then applying Langevin dynamics based on the estimated scores is because $p(\boldsymbol{x}|\boldsymbol{y})$ as an optimization target is computationally intractable. This is due to the fact that the computation of  $p(\boldsymbol{x}|\boldsymbol{y})$ is not scalable with the dataset size. If $p(\boldsymbol{x}|\boldsymbol{y})$ is accessible, conditional sampling can be accomplished by directly applying Langevin dynamics to its scores, without even using any form of score-matching.
>
> Second, a good cross-entropy loss does not guarantee a good approximation of the gradients. This can be observed in the counter example presented in Fig. 5, in which the score error grows even when cross-entropy loss decreases. This evidence thus suggests that minimizing the cross-entropy loss does not necessarily lead to a good gradient approximation.

---

> ### Author Response · Authors · 2021-11-21
> **Response to Reviewer LMQh (Part 1/3)**
>
>
> We appreciate the time and effort from the reviewer.
>
> ---
>
> **Q1.** The loss is quite noisy (see Figure 5), and the authors do not explain why. I personally think that the reason is because they approximate $p(y|x)$ with a mixture of Gaussians with a small variance and there is the same effect with smoothing as in case of the standard score matching. Probably we should start training using larger variance values and gradually decrease the variance during the training.
>
> **Q6.** Why is their loss noisy in practice? How much does the value of $\tau$ (noise variance) affect the stability of the loss and the results? Why don't the authors do the same trick with $\tau$ as with $\sigma$ (that is, they don't train several different models with different values of $\tau$)?
>
> **A1 & A6.** We would like to first clarify that the training instability shown in Fig. 5 does not result from $\tau$. Instead, it is due to the training objective. To be more specific, the unstable trend of the blue line in Fig. 5 (a) is because the score-matching loss is not explicitly minimized during training. Similarly, the unstable decreasing trend of the orange line in Fig. 5 (b) is because the cross-entropy loss is not explicitly minimized during training.
>
> The reviewer’s proposal to schedule the value of $\tau$ during training may be beneficial (i.e., gradually decreasing the variance), however, our experimental results in Table 2 have shown that the proposed method demonstrated promising results on two real-world image datasets. Therefore, we consider that the scheduling of $\tau$ is optional.
>
> ---
>
> **Q2.** I think that the motivation of the paper has too many somewhat irrelevant details. The authors compare different models, trained under different conditions, and conclude that the problem is in the loss. It seems that it was enough to say that it is difficult to approximate the gradients, so it is necessary to explicitly add them to the loss.
>
> **A2.** All of these settings are necessary to support the arguments presented in this paper.  The reasons of introducing methods (a), (b), (c), and (e) are elaborated as follows:
> - **(a) Base Method**: It is necessary to include this method as the **baseline**, since the main purpose of this paper is to improve it.
> - **(b) Scaling Method**: The scaling method has to be included as the **baseline** to highlight the effectiveness of the proposed approach. This is because the Ablated Diffusion Model (ADM) [1] achieves superior performance on several representative datasets by utilizing this method.
> - **(c) Posterior SM Method**: The posterior SM method serves as an main **motivation** for us to derive the proposed DLSM loss. Since the DLSM loss is the main contribution of this paper, it is also necessary to introduce method (c).
> - **(e) Oracle**: This method provides the optimal scores. The optimal scores are necessary since the **quantitative evaluation** requires measuring the score errors between the estimated and the optimal ones.
>
>
> [1] Diffusion Models Beat GANs on Image Synthesis, Prafulla Dhariwal et al., ArXiv 2021

---

### Decision · Program_Chairs · 2022-01-20

**Decision:**

Accept (Poster)

**Comment:**

This paper discusses an issue with decomposing a conditional generative model into an unconditional model and a separate classifier using Bayes' theorem, which is an approach that has recently received increased attention in the context of score-based generative models. It explores several alternatives for mitigating this issue, including a novel one, which is to use a different loss function to train the classifier.

Reviewers praised the writing and the way this work draws attention to an issue that is underappreciated in the community. Although several weaknesses (clarity, scale of experiments, appropriateness of baselines, missing experiments) were also highlighted in the original reviewers, all reviewers agree after discussion that the authors have adequately addressed these for the paper to be considered for acceptance. I will follow their recommendation and recommend acceptance as well.